# Information content in reflected signals during GPS Radio Occultation observations

Josep M. Aparicio[1,3], Estel Cardellach[2,3], and Hilda Rodríguez[2,3]

[1]Meteorological Research Division, Environement and Climate Change Canada (ECCC), 2121 Transcanada Hwy, Dorval, QC, Canada
[2]Institut de Ciències de l'Espai (ICE), Consejo Superior de Investigaciones Científicas (CSIC), Cerdanyola del Vallès, Spain
[3]Institut d'Estudis Espacials de Catalunya (IEEC), Cerdanyola del Vallès, Spain

*Correspondence to:* Josep M. Aparicio (Josep.Aparicio@canada.ca)

**Abstract.** The possibility of extracting useful information about the state of the lower troposphere from the surface reflections that are often detected during GPS radio occultations (GPSRO) is explored. The clarity of the reflection is quantified, and can be related to properties of the surface and the low troposphere. The reflected signal is often clear enough to show good phase coherence, and can be tracked and processed as an extension of direct non-reflected GPSRO atmospheric profiles. A profile of bending angle vs. impact parameter can be obtained for these reflected signals, characterized by impact parameters that are below the apparent horizon, and that is a continuation at low altitude of the standard non-reflected bending angle profile. If there were no reflection, these would correspond to tangent altitudes below the local surface, and in particular below the local mean sea level. A forward operator is presented, for the evaluation of the bending angle of reflected GPSRO signals, given atmospheric properties as described by a Numerical Weather Prediction system. The operator is an extension, at lower impact parameters, of standard bending angle operators, and reproduces both the direct and reflected sections of the measured profile. It can be applied to the assimilation of the reflected section of the profile as supplementary data to the direct section. Although the principle is applicable also over land, this paper is focused on ocean cases, where the topographic height of the reflecting surface, the sea level, is better known apriori.

## 1 Introduction

Signatures of the interaction of the GPS signal with the Earth's surface have been recorded during satellite-based radio occultation (GPSRO) events (Beyerle and Hocke, 2001; Beyerle et al., 2002; Pavelyev et al., 2011). Besides the usual refracted signal (hereafter named direct) at the limb, another signal is frequently detectable with a small frequency offset, corresponding to lower elevation, often among distorting phenomena of multipath propagation, superrefraction (unusually large refractivity gradient) and strong attenuation, and is not always clearly defined. Both signals merge when the tangent altitude of the direct

propagation path is at the surface. The abovementioned studies show that this is the expected behavior, and particularly the frequency offset, of a signal that would have reflected off the Earth's surface, and that was unintentionally captured by the receiver, while it was tracking the direct signal. This second signal will be hereafter named reflected.

This reflected signal has interacted with the low atmosphere and the surface, and therefore constitutes a potential source of information of their properties, in addition to the atmospheric information that is normally provided by the direct signals. In this study, a large number of recorded occultation events are analyzed for the presence of reflections, in order to characterize when reflections are to be expected, which information can be found in these reflected signals, and how this information can be used within the context of atmospheric applications, and particularly Numerical Weather Prediction (NWP).

The first task would be to be able to identify and separate the reflected component. GPS satellites emit in several regions of the L-band. The most important in the context of this work is the L1 (near 1.575 GHz, wavelength approximately 0.19 m). The signal is a very pure and coherent tone, modulated by several predictable phase shift patterns (Parkinson and Spilker (1996), Ch. 3), which are specific to each emitter. The coherence, or phase stability of the emitted signal, is of particular importance for the present purpose, or in general for GPSRO, as it allows the interpretation of phase observations as signatures of the interaction of the wave along its path, with the atmosphere, ionosphere or Earth's surface. If the emitter was unstable, this phase noise would overlap with the signatures of those interactions, complicating the use of phase as a measurement. The same happens if the interaction introduces random changes in phase. The effect of phase noise may lead to the inability to use the measurements interferometrically, or if it is large, even to loss of track by the receiver (inability to record the signal for postprocessing).

In this work we focus on the C/A Pseudo Random Noise (PRN), a 1.023 MHz binary modulation of the sinusoidal tone. Receivers collect signals from several emitters, overlapped, and apply several discriminating layers to focus into individual components. Firstly, GPS signals are emitted with right-handed circular polarization (RHCP), and GPSRO receivers expect to receive signals that are still close to RHCP. At near-normal incidence (see Figure 1) a reflection would reverse the polarization handedness, (for instance Born and Wolf (1999), Sec 1.5.3, Ulaby et al. (1986) or Zavorotny and Voronovich (2000)), and the receiver antenna would be able to discriminate a reflection. This is the case of dedicated GNSS reflectometry missions such as NASA's CyGNSS (Ruf et al., 2013). However, during an occultation, the signal has low elevation over the horizon, and a reflection does not reverse the polarization. The reflected signal is also right-handed and captured by the antenna intended to collect the direct signal.

Thanks to the characteristic modulation, receivers can decompose the received overlap into a sum of several emitters. Signals from the same emitter that have followed two propagation paths may also be identifiable as distinct (Parkinson and Spilker (1996), Ch. 2, Sec VI), provided their optical length is sufficiently different, by at least one modulation bit, which at 1.023 MHz corresponds to ∼300 m. Each identified component can then be tracked separately. During an occultation, the direct signal has low elevation over the horizon (see Figure 1). Any reflection will be grazing the surface, also at very low angular elevation, and the difference in their path length will be relatively small, of up to several tens of meters. This is large compared to the wavelength, but small compared to the length of GPS's modulation pattern of approximately 300 m. Direct and reflected paths therefore show nearly identical modulation. Similarly, their frequencies (affected by Doppler effects) are also quite

similar, separated by only few Hz, and within the usual receivers' bandwidth. Although GPS receivers are designed to separate in most cases the different signal components through hardware, either by polarization, through the Doppler shift, or by delay through PRN modulation, the specific case of the direct and reflected paths near the horizon is particularly challenging. The reflected signal is thus recorded by the receiver when it is targeting to track the direct signal.

This separation is still possible during postprocessing, as detailed below, analyzing the spectral distribution (Hocke et al., 1999) of the sum of both direct and reflected signals. Separability is possible if the spectral distributions of the direct and reflected signals are sufficiently narrow and distinct. This is indeed often the case, and as shown below, the analysis of large amounts of occultations (over 4 million events, worldwide distributed over 9 years) indicates that in a substantial fraction both direct and reflected signals are sufficiently clear to be separable, and that both still substantially maintain the coherence of the

emitted tone, which allows their use for further processing (Cardellach et al., 2008). Given that the signal phase plays a central role in GPSRO data, which is after all an interferometric procedure, the narrow spectral distribution of these signals, and thus the coherence of each, is critical to their value as a source of interferometric data.

A test of the geographical distribution clearly indicates a correlation between the clarity (coherence) of the reflected signal captured by the receiver, and qualitative properties of the surface, being particularly common over comparatively smooth

surfaces, and notably the oceans and ice in polar regions. This is the expected behavior for quasi-specular reflection, which maintains signal coherence. Land, instead, with generally a much more irregular surface, is expected to produce diffuse scatter, where signal coherence is lost. Boniface et al. (2011) showed that reflected signals in radio-occultation events could also contain geophysical information about the lowest layers of the troposphere (lowest few km), provide information about the state of the low troposphere, and are therefore potential sources of meteorological information, additional to the non-reflected signals,

which have suffered only refraction, and that are already used as observations that are inverted to profiles of atmospheric bending angle or refractivity.

This frequent clarity in the spectral separation between the direct and reflected signals implies that, in principle, it is possible to interpret their properties independently, and that both may provide useful geophysical information. This has been shown to be possible, for instance, in occultations over ice in Polar Regions (Cardellach et al., 2004), where the extracted quantity

was the height of ice above sea level, from interferometric phase data of the reflected signal. Besides the height of the surface, the reflection depends also on the refractive index of the atmosphere, mostly on the properties of the low troposphere, which is the focus of the present study. It is attempted here to extract atmospheric information that would be usable in the context of NWP, particularly to provide additional information about the temperature and moisture profile in the lowest 5 km. This study is restricted to ocean reflections, where we can assume that the height of the surface is well known a priori: the average

sea level. The sea surface itself is continuously changing due to ocean waves; however it departs vertically by at most few meters from the mean sea level. In limb-looking geometry, this departure of the actual sea surface from the mean sea surface modifies negligibly the length of the propagation path. A sea reflection in this geometry is effectively specular. Over land, topography, albeit static, is a strongly irregular surface near the tangent point, whose reflective properties are also spatially very variable. The exact vertical location of a reflection over land is in general unknown by tens or even hundreds of meters.

Besides showing less cases of clear coherent reflection, the practical use of land reflections in those cases where the signal is

coherent would require solving, besides tropospheric properties, additional unknowns related to the topographic height of the reflecting surface. Therefore, the qualitative properties of reflections over land are explored in this study, but the possible use of reflected interferometric products in NWP over land is, for this reason, considered out of scope.

Atmospheric information normally extracted from radio occultations is derived only from direct data, which has interacted only with the atmosphere, through refraction. If atmospheric information can also be extracted from ocean reflected data in at least a significant fraction of the occultations, as indicated in this study, we may infer the existence of a significant pool of data sensitive to the properties of the lower troposphere, which can be useful for applications in NWP, and which is so far unused. These would be additional to, and well collocated with, the standard radio occultation profiles.

The fact that the open ocean shows better signatures, and that these can be interpreted as atmospheric measurements, rather than a combination of atmospheric and topographic, is well suited to the needs of data acquisition for weather forecast, as the pool of available low altitude information from other sources is particularly scarce over oceans. Radio occultation has otherwise provided a very significant amount of atmospheric data worldwide, which has been of particular value over remote areas, including oceans and polar regions, e.g. Healy and Thépaut (2006); Cucurull et al. (2007); Aparicio and Deblonde (2008); Rennie (2008). These same studies of forecast impact indicate that the low troposphere is the region where direct (refracted-only) radio occultations are less accurate, and therefore where adding supplemental information would be most helpful.

Existing GPSRO receivers are designed to track the direct signal. They evaluate where to find it in the Doppler and pseudo-range space. The reflected signal is accidentally captured if it falls in a small Doppler and pseudorange neighborhood around the direct, which happens only during the deepest part of the profile. A receiver designed to search for both independently, may track the reflection at higher elevation. As detailed in Section 3.2, at low elevation angle, the reflection does not affect the phase coherence of the signal. As the elevation increases, the roughness of the surface destroys the phase coherence, and reflection becomes a diffuse scatter. Receivers that track the reflection exist, although their focus is mostly the measurement of this diffuse scatter of random phase, a signature of the sea state. Recently, two spaceborne missions specifically dedicated to collect GPS reflected signals have been launched into Low Earth Orbit: UK's TDS-1, in 2014 (Unwin et al., 2016) and NASA's CYGNSS, in 2016, (Ruf et al., 2013). ESA is also investigating a Global Navigation Satellite System (GNSS) reflectometry (GNSSR) experiment from the International Space Station, the GEROS-ISS (Wickert et al., 2016). Although these missions use GPS and GPS-like signals to infer properties of the Earth's surface, it must be noted that the technique described in this study is not the same. The main differences are, firstly, the geometry (near-nadir in these dedicated missions vs. grazing observations in this study) and the nature of the scattering. At near-nadir angles, the irregularity of the surface (topographic, ocean waves, etc) is larger than the GPS wavelength. Scattering is diffuse and incoherent. Receivers do not expect to collect a coherent electromagnetic signal. At grazing incidence angles, as in this study, the path difference caused by the irregular surface may be negligible. The receiver is identifying a signal that still maintains coherence, and where interferometry is still possible. A non-scaled sketch of both concepts is depicted in Figure 1. Table 1 also summarizes the main differences.

Another mission that it is interesting to compare against, is the active Synthetic Aperture Radar (SAR) component of the Soil Moisture Active-Passive (SMAP) mission (Piepmeier et al., 2017), which also studies the reflection on the Earth's surface

of L-band signals, in this case at 1.26 GHz. The geometry of this mission is also near-nadir, and therefore comparable with the dedicated missions just mentioned, which analyze the incoherent scatter.

This study is structured as follows. Section 2 overviews some properties of the GPS signals, that are relevant to the grazing reflections during limb observations. Section 3 explores a large sample of GPSRO events, looking for reflections. A number of qualitative properties are identifiable in the average occurrence: there is an ocean vs. land difference, according to qualitative properties of the surface, detectable seasonal cycles, and relationship with some atmospheric properties. The main identifiable trends are described in this section. The statistics of reflection clarity and occurrence indicate where and when it is to be expected to find clear coherent reflected signatures, and therefore where new interferometric information is present. Section 4 explores the separation of the reflected signal, for those cases where coherence allows it, and the creation of an interferometric profile product out of the reflected signal. This product is similar to the standard bending angle profile obtained from direct data, but supplementary to it. It is shown that these reflected profiles are very sensitive to the refractivity of the low troposphere. An operator is presented that may use this information in the context of NWP. It is an extension to a standard class of operators that already use direct radio occultation data in this context. Examples of use are provided. Section 5 comments the results obtained, and the potential lines of further exploration.

## 2   Received signals

GPS satellites produce very pure sinusoidal tones. Before emission, these tones are modulated by characteristic bitstreams. Focus here is on the C/A signal, modulated by a PRN code at 1.023 MHz, and also by a 50 Hz stream of navigation bits. The receiver hardware can decompose and demodulate the received PRNs, and we assume that the navigation stream can be obtained from either direct measurement or from ancillary sources. Also, the orbits of both emitter and receiver are assumed to be known, from the analysis of navigation data, allowing the subtraction of any trivial (straight-line propagation) orbital Doppler. If an undisturbed signal is received, demodulation and Doppler subtraction will return the signal to the original unmodulated sinusoidal carrier tone. Since the propagation has traversed the ionophere, the atmosphere, and perhaps bounced off the surface, there may be delays, multipath, or focusing/defocusing. After demodulation and orbital Doppler subtraction, the signal recorded by the receiver will show residual Doppler, amplitude variations, and albeit still narrow, a broader spectrum than the very pure tone emitted. These are the signatures of the interaction, and constitute the primary data for the analysis of atmospheric radio occultations (Kursinski et al., 1997), where they are interpreted as observations indicating the refractivity of air, at different altitudes. These are as well the primary data for the present purpose.

The profiles analyzed here are obtained from the Constellation Observing System for Ionosphere and Climate (COSMIC) mission (Anthes et al., 2000), which maintain a good signal to noise ratio at low altitude. Data from only one of the GPS frequencies are considered: L1 (1.57542 GHz). After demodulation, it is sampled at 50 Hz. COSMIC also collects signals from GPS L2 (1.22760 GHz), but these measurements are noisier, both due to the lower emission power of L2, and because it is modulated by the bitstream known as P(Y). Since this bitstream is encrypted (Parkinson and Spilker, 1996), receivers that cannot decrypt it, like COSMIC, must resort to a different and less accurate tracking algorithm. During occultations, L2

can only rarely be tracked within the lower troposphere. This is not a particular limitation, neither for the standard processing of direct non-reflected signals (Zeng et al., 2016), nor for the processing of reflected signals presented here. L2 is required to identify and quantify the effect of the ionosphere, and for such purpose it is sufficient to record and process L2 at higher altitude. Atmospheric radio occultation profiles of refracted-only data are also obtained with only L1 in the low troposphere (Zeng et al., 2016). Receivers to be deployed in the future may add the ability to extend measurements at a second frequency at lower elevation, thus to similarly capture the reflection. Notably, the civilian bitstream L2C that is being added to new emitters to the signal at the frequency L2, can be tracked with much higher accuracy than L2 P(Y), which will help future receivers to track two independent signals, at two frequencies, into the low troposphere.

The measured record is thus a 1-dimensional (in time) stream of signal phase and amplitude, already demodulated and corrected from orbital Doppler. Following the principle of the radio-hologram (Hocke et al., 1999), another reference signal is created, intended to capture the main, low frequency evolution of the phase, but without fluctuations in amplitude, or in high frequency in phase. The received signal is then counter-rotated by this reference, leaving it in a narrow frequency band. The target, in particular, is to not exceed one half of the sampling frequency, and to avoid aliasing with the 50 Hz sampling bandwidth of COSMIC receivers.

The remaining activity is analyzed with a sliding Fourier Transform. The new representation is obviously redundant, as it is a 2-dimensional representation (time, frequency) of the initial 1-dimensional (time) stream of data. In exchange, it allows a better representation of the departure of the recorded signal from a pure tone. Besides the tone corresponding to the direct signal, a supplementary tone may be identifiable, with a specific frequency offset structure that suggests a reflection at the surface (Beyerle and Hocke, 2001). One or both of the ordinary and supplementary tones may be broadened, or even split. Figure 2 shows an example of a radiohologram that presents a reflection. The broadening and splitting (multipath) are common, and suggest a complex refractivity structure in the low troposphere.

The direct propagation paths probe the low troposphere only when the tangent altitude is in the low troposphere. Reflected paths, instead, always probe the entire atmosphere, including the lower section, albeit at varying incidence angles. The accuracy of an atmospheric profile retrieved exclusively from direct propagation, at any given altitude, is limited by the accumulation of any error or distortion at all upper altitudes. The supplementary reflected tone may also present complex structure, especially at very low incidence, where it may even be difficult to differentiate from the direct path. However, reflected signals are also detectable far from this complex frequency broadening, when both paths are narrow and distinct. Whereas complex structure in the direct path makes it difficult to probe the properties of the lower troposphere, the reflected tone may present sections that are less broadened in frequency, offering additional opportunities to provide information about these low layers.

## 3   General description of reflections in radio occultation

This section first briefly summarizes, and then extends, the preliminary studies presented in Cardellach et al. (2008, 2009), a methodology applied to identify and quantify reflected signals in occultation profiles, to study the geographic and seasonal distribution of the occurrence of reflections during occultations, and to provide some quality indicators.

The identification of the presence of reflected signals is based on Support Vector Machines (SVM), a supervised computer learning method (Joachims, 2002). The algorithm allows the training of a computer for automatic identification of target features. The algorithm is applied here to the 2D radio-holographic representations of the event, an example of which is shown in Figure 2. The details of the implementation for this particular process are compiled in Cardellach et al. (2009). The SVM is trained to identify the presence of the secondary (reflected) tone in the radio-hologram, and to provide a quantitative indicator of its clarity. The output of the SVM is a real-valued scalar, which is related to the likelihood of presence of a reflection signature: an above-noise, compact feature in the radio-hologram, with the appropriate shape and location in the time-frequency space to agree with the expected properties of a reflection. The scalar presents positive values for likely-reflection, and negative values for a lack of it, and is normalized to provide values beyond $\pm 1$ when the algorithm has a very large confidence in the resulting classification, and between -1 and +1 when the system shows only some moderate confidence.

The SVM algorithm was trained (Cardellach et al., 2009) with a set of 6468 occultations, selected by visual inspection of the radio-hologram as representing events of clear reflection, and clear lack of reflection. Once trained, the SVM was validated with an independent set of GPSRO occultations observed with the COSMIC mission, and consisting in 3350 events in February 2007, and 2257 in November 2008, also visually inspected and manually flagged. The validation determined (Cardellach and Oliveras, 2016) that the percentage of error (false positives or negatives) of the SVM-based classification is acceptable when the absolute value of the SVM output is greater than 0.25 (97.81% success for the February 2007 sub-set, and 99.47% for November 2008).

This trained SVM is then able to automatically flag other occultations. It has been used here to study the statistics of reflection events in a much more extended set of occultations, which extensively includes several years of COSMIC data. This set represents over 4 million radio-occultations, globally distributed and broadly covering the entire Earth, all seasons and local times, from the beginning of the COSMIC mission in 2006, to mid 2015.

For the purpose of this study, radio occultation events from this list of COSMIC occultations, whose SVM output value is greater than 0.25 are identified as reflections. Given the very large success rate of classification using this threshold (low ratio of false positives), there would be little gain in classification accuracy with a larger SVM threshold. Instead, it would greatly reduce the number of occultation events identified as showing a reflection, and limit the usefulness of these as a potential source of data. The fraction of GPSRO events that present reflection is a substantial fraction of the total, and shows several clear patterns of variability, mainly according to the type of surface, ocean or land, and by latitude. It is found that $\sim$50% of the GPSRO events in this dataset over extra-tropical oceans show a reflected signal, and $\sim$70% in near-polar oceans. Reflections over non-polar land occur rarely, up to 13% of the total land GPSRO cases at latitudes $\leq 70°$. Over polar land or ice regions, the number increases significantly, up to 52%. According to these results, reflections are more likely over some surfaces, but might occur over any sort of surface (ocean, ice, land). A summary is compiled in Table 2.

## 3.1 Land events

The coherent reflected signal that the SVM is trying to identify, which presents power concentrated in a narrow frequency band, corresponds to a specular reflection. The land surface appears to be less prone to this kind of interaction with the signal.

We must remind here that the GPSRO receiver and the classification of the radiohologram, identify coherent signals. Although soil moisture or vegetation may also be reflective in the L-band, irregular surfaces destroy coherence, and lead to non-specular diffuse scatter. Nevertheless, coherent reflection still appears to take place over land in a significant fraction of events, more frequently over areas known to be smooth, both in topography and when they are free of thick vegetation (see Figure 3). Among others, could be mentioned central Asia, near the Caspian and Aral seas, or certain areas of the central US and Canadian prairies. Reflections appear to be particularly common over continental ice (Greenland, Antarctica). Given the electromagnetic band at which GPS works (L-band, wavelength $\sim 0.19$m) and the very slant geometry of these observations, the general lack of reflections over rugged terrain, and where thick structures of vegetation canopy are present, could be expected.

The presence in desert areas, generally not particularly reflective, of some of the events that the SVM classifies as reflections, suggests that we should not discard that these events include as well some cases of non-reflective interaction near the surface, such as a mirage. For the purpose of tracking this signal, and extracting information of the refractive index of the low troposphere, the difference between both is negligible: there is an effective quasi-specular surface at the ground, or very near to it. We will hereafter use the label *reflection* meaning that the signal propagates at a very low elevation angle until the surface, and coherently proceedings forward to the receiver. Despite the low percentage of presence of a reflected signal over land surfaces, seasonal patterns can be noticed in the areas where reflections are more common. This is shown in Supplementary Movie 1, which presents the median SVM output value over land, in bi-monthly sections over the seasons, accumulating all 9 years of data.

Some of the seasonal patterns are found in near-polar land areas, where there is a winter cover of snow and ice, and little difference is to be expected during this period with respect to the behavior over neighboring ocean ice (northern coasts of Alaska, Canada, and Russia). Additional patterns are found (see Supplementary Movie 1) over certain regions that are topographically flat, consistent with the seasonal thinning and thickening of deciduous vegetation. Examples are the Pampas region in Argentina, a steppe, and the Southern Great Plains in the US. In all these, more reflections appear during the local winter, as should be expected if reflection takes place at the moist or snow-covered ground, or is otherwise diffusely scattered by vegetation during the local growing season.

## 3.2 Ocean events

As mentioned in Table 2, the percentage of captured ocean reflected signals strongly depends on latitude. Figure 4 displays the geographical distribution of the fraction of ocean events that show reflections, for two bi-monthly periods of the year: January-February, and July-August, both accumulated over the ensemble of events, which spans 2006-2015. These geographical distributions also show seasonal patterns, which are particularly prominent where larger gradients of reflection percentage appear, in the midlatitudes of both northern and southern hemispheres. The tropical oceans always present much lower percentage of reflections than mid-latitude ones. Comparison of the two panels in Figure 4, shows that this tropical region with less reflections is shifted southwards during austral summer, and northwards during boreal summer. The maps also show certain interesting geographic patterns: when compared against their latitude band, certain ocean regions show a systematically larger or smaller fraction. Areas that show larger fraction are, for instance, in the Pacific Ocean, the northeast (California) and

southeast (Chile), and in the Atlantic Ocean, the southeast (Namibia), northwest (New Foundland), and the northwest African coast.

The ocean surface is much less rugged than land, and at these very slant geometries and for the L1 wavelength, it should behave as a highly reflecting smooth surface (Ulaby et al., 1986). The fraction of events over open ocean is indeed larger than over land in general, with the exception of ice-covered land. However, there is still a significant portion of the ocean GPSRO events that do not present clear reflected signals. As detailed below, it is here explored the possibility of this lower rate of coherent reflections in the tropics may be caused by instrumental limitations, by the sea surface itself, or by the atmosphere.

Concerning instrumental limitations, these signatures appear at the deepest part of the occultation, when the direct radio link scans the lower troposphere, and where the tracking algorithms might already have difficulties. This may explain a percentage of missing reflections in ocean GPSRO data, as well as the difference found between the percentage of reflections in rising and setting occultations (almost a factor of two). The depth that is reached by the GPSRO profile may be linked to the opportunity of capturing a reflection: if an event does not reach the low troposphere, it may as well be unable to collect the reflected branch of the signal. This hypothesis has been checked through the statistical distributions shown in Figures 5 and 6, and found to be only partially true. The ocean events with likely presence of reflection (high SVM output values) do tend to reach the lowest layers of the troposphere, especially the lowest 2 km (Figure 5), which is consistent with the hypothesis. However, as shown in Figure 6, the ocean GPSRO events that do reach the low troposphere still may or may not present reflected signals. In Figure 6, the distribution is still well populated for negative SVM output values (no clear reflection). Therefore, reaching the low troposphere during an occultation over the ocean seems to be a favorable but not a sufficient condition to capture a reflection.

The latitude profile of the percentage of reflections is shown in the upper panel of Figure 7. The profiles are evaluated over bi-monthly blocks through the seasonal cycle, accumulated over the entire study period. It shows that the pattern of lower fraction of reflections in the tropics and lower midlatitudes displaces smoothly in latitude following the seasons, and that the fraction of reflections at latitude $> 45°$ is always large and about 70%. This seasonally-driven meridional shift of the reflection percentage can also be clearly seen in the maps in Figure 4, as well as in Supplementary Movie 2, which shows the median SVM output value over ocean, evaluated over the seasonal cycle, in bi-monthly sections, accumulating all 9 years of data.

The fraction of events that presents reflections shows a large meridional gradient in the bands at latitudes between $40°$-$50°$, both north and south. In these bands the fraction of reflections changes quickly from sparse (20% in the tropics) to frequent (70% circumpolar). Due to this gradient, the local seasonal pattern presents in these two bands the largest seasonal amplitude. The lower panel of Figure 7 shows the monthly fraction of reflections in each of these two bands, over 2006-2014, which also underscores this seasonal pattern, as well as the opposite phase between both hemispheres. These patterns, consistently associated with geographic and seasonal features, do not suggest any direct relationship between this modulation and any instrumental problem. Therefore, the core modulation of the lower fraction of GPSRO reflections in tropical oceans, or during the respective warmer season of midlatitude oceans, must be searched in environmental phenomena, either atmospheric or oceanic.

Some indications about the associated geophysical phenomena can be found empirically, correlating certain weather parameters at the location/time of the event against the presence of reflection. Fields from the ECMWF ERA Interim analysis (Dee

et al., 2011) were used here to compare against several weather or environmental parameters. Among the variables considered, those that depend on height, such as air temperature or moisture, have been averaged over the lowest 10 km. The correlations presented in the following are all computed using monthly averages of these variables, evaluated on cells of $10° \times 10°$ over the oceans, with the reflection fraction compared against them. The number of pairs to correlate is therefore the number of

ocean cells times the number of months considered. This clustering of 4 million events into several thousand geographic and seasonal groups simplifies the correlation, without modifying significantly the statistical results. This clustering produces of course some smoothing of the more variable atmospheric fields towards their average, reducing the extremes. Since the modulation of reflections affects a very large percentage of events, if it is mediated by an environmental variable, the modulation must be identifiable also in the core of the distribution of this environmental variable, and not exclusively at its extremes. The

study of these clustered averages is thus still meaningful.

We might think that, as for land reflections, the surface roughness might play some role. At these grazing incidence angles, even the largest ocean waves are very small compared to the vertical scale that determines signal coherence, the Fresnel diameter (which is $> 100m$). The difference in path length $d$ that would be associated to waves of height $H$ in the surface is $d = 2H \sin(e)$, with $e$ the elevation angle of either the emitter or receiver, seen from the reflection location. $e$ is less than $1°$,

and $d$ is thus a small fraction of the GPS wavelength for all realistic wave heights. At this incidence angle, the sea is effectively a very smooth surface, as sensed by at GPS wavelength. Reflection will be specular, and will not destroy phase coherence. The surface roughness due to waves cannot be the prime driver that determines the presence or lack of reflected signals. Indeed, the correlation between percentage of reflections and significant wave height happens to be very low, at $r = 0.04$, which confirms the above estimation that the roughness produced by waves is not sufficient to affect the coherence of the reflection.

On the other hand, the correlation of the fraction of reflections against the sea surface temperature (SST) is strong and negative (r = -0.80): warmer sea surface temperatures are associated to a low fraction of reflections. The relationship between reflected signals and SST was also visible in Figure 4. The above mentioned features along the West coast of North and South America, West South Africa, Northwest Africa, West Australia and North-Eastern North America are all regions of relatively cold surface water. Similarly, regions of typically warmer water in the Coral Sea, Caribbean, East Africa and the Arabian Sea

are also regions presenting comparatively less reflections. These features present as well (Supplementary Movie 2) seasonal variations consistent with sea surface temperature.

The dependence of the electromagnetic reflectivity with temperature at the air to sea water interface was explored as a potential cause, but was ruled out. Despite being true that the Fresnel reflection coefficients present some dependence on the temperature of the surface (Ulaby et al., 1986), the coefficients for the co-polar component of circularly polarized signals

(RHCP incident, RHCP reflected) do not change significantly with temperature at the grazing incidence angles of these observations: there is less than 0.1% variation over temperatures between $1°$ and $20°$ C, at incidence angles greater than $80°$ (i.e. low elevation).

As a consequence, the modulation of the occurrence of ocean reflected signals in GPSRO events cannot be directly linked to the surface itself, its roughness or its temperature. Instead, it should be linked to the different atmospheric conditions that take

place above the water surface, but associated to its temperature. Cross-correlations with other atmospheric variables were also

performed and are shown in Table 3. For instance, Figure 8 shows the percentage of reflections against the column of water vapor (CWV). Most of these variables also show a meridional and seasonal dependence. The correlation, however, may be stronger or weaker than that of SST, which may help to identify the primary cause of clear or unclear reflections on ocean. The presence of coherent reflected signals also appears to correlate significantly with the column of water vapor, and the vertical

integral of total energy, in all cases negatively, with correlation coefficients close to -0.8. The next variables, by strength of the correlation, are the temperature in the low troposphere and water vapor pressure (correlation r around -0.7). However, the relative humidity does not seem to play a central role (r=0.26). As mentioned above, the significant sea wave height is nearly uncorrelated (r=0.04), although wind speed over sea has a moderate positive correlation (0.43). This was somewhat unexpected, as stronger winds correspond to rougher sea surfaces, which could seem to link to less chances of coherent reflections. The fact

that the height of sea waves does not correlate with the presence of reflected signals, added to the fact that the wind-reflection correlation is positive (there are more reflections with stronger winds), might indicate that wind itself is not the driving element either, but it would point to atmospheric conditions that tend to occur associated with wind.

A parameter that presents a high and positive correlation with the presence of reflected signals is the difference between the environmental temperature lapse rate ($\Gamma$) and the saturated adiabatic lapse rate ($\Gamma_s$), indicative of atmospheric stability.

These conditions indicate that reflections appear more frequently with a stable layering of the atmosphere. The moderately positive correlation with wind, which mixes air, and with stable layering, together with the large negative correlation with surface temperature, which weakens the stability, suggests that a likely root cause of the modulation of events that present a reflection may be the horizontal homogeneity of the atmosphere. The signal is propagating horizontally through the low troposphere, which can be optically very heterogeneous (strongly variable refractivity). The correlations found are compatible

with coherent reflections appearing when the atmosphere is well stratified near the tangent point. It is interesting to note that these stable conditions are also related to the appearance of propagation ducting (von Engeln and Teixeira, 2004), that is, cases of large refractivity gradients that trap a signal, or superrefraction, which show a very similar spatial and seasonal distribution (Xie et al., 2010).

The relationship between the preservation of coherence and stability suggests that the physical relationship involved is the

homogeneity/irregularity of the field of refractivity. The signal is traversing the atmosphere quasi-horizontally, through some 200 km of low troposphere. The atmosphere along this path may be optically irregular, presenting strong variations of refractivity, distorting the propagation of the signal. This was tested verifying if the atmosphere was optically homogeneous in the vicinity of the occultation, in all ocean events (about 10000 occultations) over a period of 3 weeks (1-21 July, 2015). The standard deviation of the refractivity at a given altitude, within 200 km around the occultation, was taken as a measure of

optical heterogeneity. Since atmospheric refractivity is mostly modulated by moisture whenever the amount of water vapor is significant, this is primarily a measure of irregularity in the field of water vapor. The horizontal heterogeneity in the refractivity was evaluated from the background fields of Environment Canada's operational short-term global forecasts (Charron et al., 2012), whose grid resolution is 25 km. The correlation of coherent reflections against this parameter indicates that these are indeed less frequent with more heterogeneity. Correlation was performed against heterogeneity at different altitudes. This rela-

tionship is negative, and is stronger (r=-0.64) against the optical heterogeneity at altitudes between 1.5-3 km. This corresponds

approximately to the height of the atmospheric boundary layer, and is a physically expectable behavior, since there are indeed particularly strong refractivity gradients at that altitude (Guo et al., 2011).

To summarize, ocean reflection events are frequent, but their presence appears to be inhibited under some atmospheric conditions, the less favorable being warmer surface temperature, and weak winds. Further exploration of this modulation will be required in future studies, although the ensemble of trends identified suggests that the physical mechanism may be related to the optical heterogeneity of the refractivity of air, which is mostly determined by the heterogeneity of the water vapor field. The layer at 1.5-3 km altitude appears to be the most involved in this link between the properties of the atmosphere and the presence of a reflection during an occultation.

## 4  Numerical Weather Prediction value

The previous section presented the observed fraction of occultations that show a reflection signature. Over ocean, the fraction is large enough to constitute a large pool of available events, and therefore worth processing further.

Cardellach et al. (2008) noticed that radio-occultation events that have reflected signals tend to correspond to cases where the extracted atmospheric profile of refractivity, obtained in that case exclusively from direct signals, agrees better with the values estimated from an NWP model, than cases without clear reflected signals, as tested against the European Centre for Medium Range Weather Forecast (ECMWF). The presence of the reflection was there used as a flag, signaling a particular statistical behavior of the measured data within a subset of events, but no additional measurements were extracted. That study was then extended to nearly 170,000 GPSRO events over oceans in Cardellach and Oliveras (2016). The results of this second analysis confirmed a better agreement between direct data and the background field when reflections are present: both the bias and the RMS difference between the measured refractivity and the NWP background field are significantly smaller in the lower 10 km of the troposphere. These results were also consistent with the study by Healy (2015). Although this does not mean that the data are of better quality, it does mean that the presence of reflection is at a minimum a marker for data that show better observation minus background (OMB) agreement than the average GPSRO event. It is interesting to note here, that the reflection flag, either a qualitative present/absent, or the quantitative SVM output value, stem from the observation, and that the knowledge that the direct profile of a given occultation is expected a priori to show lower OMB difference than an average occultation is already information that is additional to the direct profile itself.

In those studies, the reflected portion of the signal was only an indicator for a distinct expected statistical distribution of the direct profile, and was not used in the construction of an output data product. We try here to produce supplementary assimilable data to the standard output profile, using phase and amplitude data extracted from the reflection signature, and producing extended profiles with additional bending angle data.

Both the height of the reflective layer and the properties of the low troposphere have an effect on the optical path length, and on the relationship between path lengths at different angles of incidence. Therefore, as a first step, and in order to simplify the interpretation of the data, we will restrict to the case where we know a priori the topographic height of the reflecting surface with good accuracy (few meters), which in this study means the open ocean, whose topography we will assume to be equal

to the mean sea level, as determined by the reference equipotential surface of a geoid model (NIMA, 2000). If finer precision of the reference surface was needed, tidal models could also be added. In addition, there is some possibility that the reflection may not take place at sea level, for instance at an elevated duct, or in land, if the occultation is near the coast. It is here assumed that these are best handled through quality check procedures.

## 4.1 Separation of the reflected signal

Occultation events that present high SVM output values correspond to profiles with a clear reflected feature, such as the one that was shown in the radio-hologram in Figure 2. In these radio-holograms, the reference signal is a smoothed version of the total signal, which closely follows the direct radio link. The reflections follow a very similar pattern (see Beyerle et al. (2002) for a detailed and quantitative study). The occultation is dominated by the motion of the receiver, which as a LEO is faster than the emitters. The reflection, presenting a lower elevation than the direct path, forms a different angle with the orbital speed of the LEO, and is the cause of the frequency offset between both signals. Both paths are equal, and thus their Doppler shifts, when the direct path has its tangent point at the surface.

A method based on the identification and isolation of the reflected branch in the spectral domain of these clear cases was implemented for this study. This is done with a 2D cross-correlation process between the radio-holographic image and a *template* 2D image. The template was built as an average case from events that were very clear, and that follow the expected Doppler shift between the direct and reflected signal, if this has bounced at the surface. The frequency offset depends mostly on the orbit of the receiver, and is very similar for all COSMIC satellites. A mirror transformation in time is applied for rising occultations. The cross-correlation identifies the time location of the reflection event within the occultation, and the time section where it is identifiable. Within this interval, the complex spectral content of the radio hologram is set to noise level for frequencies higher than $f > -5$ Hz. This frequency margin is selected to avoid contamination from the direct signal, which can be noisy, with wide frequency spread, within the section of the GPSRO event in the low troposphere. This selection, within the radiohologram, of spectral content from the reflected signal, is illustrated in Figure 9. This content is intended to include signal that is clearly part of the reflected signal, but not necessarily all of it. Some of the reflected power, especially near merging, has a frequency within less than 5 Hz of the reference signal. The receivers in COSMIC sample at 50 Hz, thus a signal will present aliasing if it is sampled at more than 25 Hz from the reference. In Figure 2, the reflected signal is identifiable at more than 25 Hz from the reference, but due to the sampling frequency appears aliased, around $f \sim +25$ Hz, at 46-48s. It is not infrequent to find reflections visually identifiable at more than 50 and 75 Hz from the reference, thus crossing the radiohologram several times. We focus here in only the last 25 Hz before merging with the direct signal. All these aliased signatures are here masked, and are not extracted.

An inverse Fourier Transform of this filtered complex spectrogram, results in a time series of complex phasors (delay/amplitude), which are identified as the reflected signal. This interferometrically obtained reflected signal presents a frequency $f_I = f_R - f_D$, where $f_D$ is the Doppler frequency of the reference direct link, and $f_R$ the one of the reflected signal. Let us remember here that the reference $f_D$ is known, as the interferogram was built as the interference beat the input signal against this reference, see Section 2. The Doppler frequency $f_R(t) = f_I(t) + f_D(t)$ of the reflected signal can be processed to bending angle and

impact parameter by means of standard equations (Kursinski et al., 1997), normally used for profiles of direct signals. An example of inversion to bending and impact parameter is shown in Figure 10. The reflected data leads to a supplementary section of the bending profile, at low impact parameter. As part of the 5Hz mask mentioned, the curve does not show the part of the reflected section that is near the direct profile, where the direct signal is tangent to the surface. In principle, both should

merge, although it is difficult there to separate the fractions of signal that are direct and reflected. Instead, reflected data that are far from this tangent path, and which can be clearly separated from the direct path, does provide a profile section. In time, the data corresponding to this section was received before the direct path reached the low troposphere. In this sense, reflected data may be present even if the tracking of the direct path is later lost. The precision of the inversion of the interferometric, and holographically extracted signal, to Doppler and then bending vs impact parameter, is very high and will be the subject of

further studies.

## 4.2 Forward operator of the bending angle

Let us consider an occultation passing through a planetary atmosphere. We will assume a quasi-spherical planet. Locally, the shape of the atmosphere will be described by a tangent ellipsoid, which observed along a given azimuth, reduces to an osculating circle of radius $R$. It will be assumed that the propagation can be described sufficiently with geometrical optics,

or that through preprocessing (such as backpropagation), the list of measured phase/amplitude values has been reduced to an equivalent that is free of diffraction Kursinski et al. (2000).

We will also assume that the atmosphere has a refractive index field $n$ that is locally spherically symmetric $n(r)$, with $r$ the distance to the local center of curvature, that $n$ never deviates largely from the vacuum value $n = 1$, and tends to it at large $r$. We also assume that the index of refraction mostly follows the trend $dn/dr < 0$, but allowing the presence of some

exceptions, which may be caused for instance by a temperature inversion layer, or the presence of a moist layer, as well as in the ionosphere.

Let $a$ be the impact parameter of a propagation ray with respect to the local center of curvature. If the refractivity field is spherically symmetric, $a$ will be conserved along the path. Let $a_S = n(R) \cdot R$ be the impact parameter of a hypothetical ray whose tangent point is exactly at the surface. A signal that propagates without ever reaching the surface, will present $a > a_S$.

These are subject only to refraction, and we will name them as "direct rays." This is the standard situation (Kursinski et al., 1997): the ray reaches a minimum distance $r_t$, at the tangent point, where $a = n(r_t) \cdot r_t$, and proceeds away from the center of curvature. The bending suffered by refraction during its propagation will be:

$$\alpha_D(a) = -2a \int\limits_{r_t}^{\infty} \frac{d\ln n/dr}{\sqrt{n^2 r^2 - a^2}} dr \tag{1}$$

The integral represents the gradual accumulation of refractive bending, inwards from $r = \infty$ to the tangent point $r_t$, and again

outwards from $r_t$ to $r = \infty$, thus the factor 2. Since the quantity $\frac{dn}{dr}$ is most often negative, bending is generally curved towards the planet (net positive $\alpha_D$). Of course, in cases where $\frac{dn}{dr} > 0$, as may occasionally happen in the low troposphere, and is normal in the ionosphere, the ray may be curved, locally, away from the planet.

A signal with impact parameter smaller than $a_S$ will reach the surface and, if the surface quality is appropriate, may reflect there. We will name these as "reflected rays". This is a physically different phenomenon from refraction, which must be included in the evaluation of the bending angle of the ray as a function of the field of refractive index. The ray differs in two ways from the direct path described in Equation (1).

Firstly, a reflection takes place exactly at the surface: $r = R$. The ray arrives there with a grazing incidence angle $\alpha_G = \frac{\pi}{2} - i$ that is not yet zero, with $i$ the incidence angle at the surface. By the Snell's law of refraction, the incidence at the surface follows $R \cdot n(R) \cdot \sin i = a$, and therefore:

$$\alpha_G = \arccos(\frac{a}{a_S}) \tag{2}$$

Due to reflection, the direction of the ray changes at the surface, by twice the elevation angle, away from the surface.

Secondly, the integral does not extend from infinity to the tangent point. This tangent point is never reached, and the gradual refractive bending instead extends from $r = \infty$ to $r = R$ (instead of $r = r_t$), and equivalently for the outgoing portion of the ray. As a result, the expression of the bending of this "reflected" ray is:

$$\alpha_R(a) = -2a \int\limits_R^\infty \frac{d\ln n/dr}{\sqrt{n^2 r^2 - a^2}} dr - 2\arccos(\frac{a}{a_S}) \tag{3}$$

Equation (3) shows bending to be partially caused by refraction (first term), associated to the atmosphere, which is generally
positive, towards the surface, and partially by reflection (second term), which if present is always negative, away from the surface. For rays with low elevation angle ($a$ only slightly smaller than $a_S$), the atmospheric bending dominates, with net positive bending. At progressively smaller impact parameters, the ray will reach the surface at larger elevation angles. The second term grows, reducing the bending, and leading eventually to negative total bending angles. The elevation angle is of course only defined for $a \leq a_S$, as for $a > a_S$ the ray never touches the surface. In general, and unlike in Equation (1), the lower
extreme of the integral does not fulfill the tangent condition. That is, it is in general *not true* that $a = n(R) \cdot R$ for reflected rays. Equations (1) and (3) represent two branches of the bending profile, respectively for $a \geq a_S$ and $a < a_S$. These two branches join continuously at $a = a_S$, where it is true that $a = n(R) \cdot R$, and $\alpha_G = 0$.

The extended reflection branch shows a qualitative difference against the direct branch: bending normally decreases at lower impact parameter. In the purely refractive case, this is unusual, around a strong refractivity gradient. A forward operator that
does not consider the possibility of reflection, and faced to an observation where bending decreases quickly near the surface, will therefore be forced to attribute it to a strong refractivity gradient near the surface. These are known to be moderately common (von Engeln and Teixeira, 2004), although the operator should be able to estimate if a given event is a surface reflection, which will be most often the case, rather than an atmospheric feature.

In the zoomed plot of Figure 10, the observed reflected section of the profile is compared against the calculated reflection,
using several realistic estimations of the refractivity profile, applying Equation (3). The estimates of refractivity include the profile determined exclusively from direct data, an NWP-estimated profile (an ECMWF profile as provided by COSMIC), and a 1DVar inversion using direct-only data. The difference between these is well above the precision of the reflected bending profile, and therefore the reflected bending is sensitive to their difference well above noise level.

Interestingly, the slope of the reflected profile is very steep, when compared with the direct profile. Therefore, whereas several candidate profiles of refractivity, as seen in direct $\alpha$ versus $a$ space would differ mostly in bending, the reflected data differ most in impact parameter space. The cause of this slope is the second term of Equation (3), which depends on the apparent elevation of the horizon $a_S = n(R) \cdot R$. Indeed, the reflected curve sharply constrains observationally this value, directly related to the refractivity at the surface. It is to be noted that the direct profile may measure the atmosphere, perhaps down to the surface if conditions are appropriate. However, the fact of having reached the surface, and therefore establishing that the profile is complete, is not determined observationally from direct data alone. Instead, for a direct profile, it is determined by our independent knowledge of the shape of the Earth. This supplementary reflected data, instead, offers an observational determination of the completeness of the direct profile.

A notable feature of both Equations (1) and (3) is the Abel kernel:

$$K(r) = \begin{cases} \frac{1}{\sqrt{n(r)^2 r^2 - a^2}} & \text{if } n(r)r > a \\ 0 & \text{if } n(r)r < a \end{cases} \tag{4}$$

This kernel is singular but integrable, and represents the exposure of a given ray, of impact parameter $a$, to properties of different layers of the atmosphere, at a distance $r$ from the center of curvature, and therefore the sensitivity of that ray to the properties of each layer. In the case of bending, the property to which the ray is exposed at each layer is the relative refractivity gradient $d\ln n/dr$. For direct, non-reflected propagation paths, this dependency of the properties of the bending with respect to the atmosphere, is very strong in a narrow range near the respective tangent altitudes, and zero below. This narrow and very distinct nature of the weighting kernel functions $K(r)$ for different propagation paths allows well determined and well vertically resolved inversions to profiles of local quantities, produced exclusively from direct data, provided a wide sample of propagation paths is available.

On the contrary, since the integral in Equation (3) does not extend until the tangent point, reflected paths are qualitatively different: they are dependent on the entire atmosphere, and do not show any narrow concentration of weight. Although slightly different, the kernels $K(r)$ for reflected paths are all very similar. An example comparing the kernels of direct paths, and the more uniform kernels of reflected data, is shown in Figure 11. The reflected kernels always include the entire atmosphere, and are always sensitive to the low troposphere. Among the direct paths, only a few are sensitive to the low troposphere, and some of the lowest may be missing. The reflected kernels have therefore the potential ability to fill a section where direct data are not sufficiently sensitive.

Although we may expect to extract some new data, additional to the direct GPSRO, which contributes to our understanding of the atmospheric profile, and especially of the low atmosphere (Boniface et al., 2011), the similarity of the weighting functions of the reflected data implies that we cannot expect, exclusively from reflected data, the high vertical resolution that is possible with direct-only data. This more limited ability to provide vertical resolution also occurs with mountain or airborne occultations (Zuffada et al., 1999), and surface meteorological GPS observations (Bevis et al., 1992).

## 4.3 Sensitivity of reflected data under ducting

The reflected radio links, like the direct ones, also suffer atmospheric bending and each link can be associated to an impact parameter. Therefore, as for direct paths, each reflected path (in a sufficiently symmetric atmosphere) can be associated to one value of bending angle and one value of impact parameter, and it is possible to build the reflected bending-impact parameter profile $\alpha_R(a)$ of a given radio occultation. In Section 4.2, it was concluded that the $\alpha_R(a)$ profile of reflected signals contains information of the lower tropospheric layers, and can be expressed by Equation 3. It was also concluded that it could contribute to determine the atmospheric solution close to the surface.

To illustrate the content of information of these reflected bending-impact profiles, an example is shown in Figure 12. A family of simple synthetic refractivity profiles is displayed on the left panel, presenting conditions close to and beyond ducting, and represent a simple case known to be difficult for a profile of direct data only (Xie et al., 2010). They are all generated from a common function, a simple exponential with altitude. An additional more refractive layer is added at low altitude (lowest 2 km), similar to the structure often found in the atmospheric boundary layer (ABL), and associated to the water vapor content, normally higher in the ABL than in the free troposphere. The additional refractivity is here shaped as an error function of several amplitudes, leading to a vertical gradient that may be strong. When this gradient is sufficiently large, the ensemble of the direct propagation rays becomes insensitive to the properties of the strong gradient, because no ray has its tangent point, the most sensitive part of the ray, within this layer. The right panel in Figure 12 shows the relationship between the reflected ray's bending and the impact parameter of each of the profiles, as given by Equation 3. Note that the right panel is a zoom showing only the reflected profile (see Figure 10 for an example with real data). It is clear that the reflected impact parameter changes distinctively with variations of the refractivity below the strong gradient (here represented with the surface refractivity value). The shape of the relationship between bending and impact parameter also changes, although less distinctively. The bending angle vs impact parameter relationship of the reflected profile is clearly sensitive to the ducting layer, and within the precision of the retrieved profiles. Although the reflected data are not resolving the internal structure of this layer, they are resolving the net step of that layer (i.e. the refractivity increment across the layer), which cannot be resolved with direct data alone.

Furthermore, regions such as SE Pacific, where superrefractive layers are common, therefore where standard GPSRO presents a weakness resolving the structure of the low troposphere, present a particularly large fraction of reflections.

## 5 Conclusions

From the descriptive analysis of a large collection of over 4 million occultations, and representing approximately 9 years of COSMIC data, we conclude that

1. it is possible to automatically detect reflected signals in GPSRO events;

2. these can happen on every type of Earth surface. Globally, about half of the occultations present a reflection signature. They are more frequent over ocean, ice, and smooth land areas, with up to 70% occurrence in medium to high latitude oceans;

3. occurrence of GPSRO reflections over land presents seasonal and geographic patterns that correlate with the smoothness of the surface, and particularly when there is a snow or ice cover;

4. occurrence of GPSRO reflections over ocean presents seasonal and geographic patterns. They do not depend on the sea wave height conditions, and instead anticorrelate with the ocean temperature.

5. the atmospheric conditions more favorable for GPSRO reflections to occur, correspond to a stable and well stratified atmosphere, with low energy (vertical integral) and water vapor, irrespective of its relative humidity.

6. strong reflected signals in a GPSRO event correlates with direct signals that have good apriori agreement with NWP.

Further studies are encouraged to determine the physical mechanism that mediates the above mentioned modulation of the clarity of reflections over ocean, although the ensemble of trends identified suggests that it is probably related to the optical homogeneity of the low troposphere, inside the occultation region.

The reflected signals frequently present a narrow spectral distribution, sufficient to be considered coherent, to be separable from the direct signal, and interpreted in terms of impact parameter and bending angle. This indicates that a large pool of bending angle data exists, which is not being used operationally, and that is additional to the standard GPSRO profiles. This pool of data is sensitive to both the refractivity of the atmosphere, and the height of the surface. If by geographic selection we limit to ocean events (known height), the reflection signature will then depend only on the atmospheric refractivity. This bending data can be simulated with an extension of standard bending angle observation operators, and is therefore usable in the context of NWP. Future work will be dedicated to practical use inside an NWP system. The dependence weight of these data is concentrated in the low troposphere, but does not have the characteristic peak of weight of direct data. Although the reflected section of the profile contains less independent information than the direct section, it contains a few unique capabilities. Since it is complete, it can provide constraints to fill the voids in a direct profile. It provides sensitivity to the interior of a ducting layers, particularly to its net step in refractivity across the ducting, and offers access to observationally estimate the atmospheric refractivity at the surface.

The interpretation of those data is facilitated when the GPSRO reflection takes place at a known altitude, and particularly the sea surface. On the other hand, reflections are also seen in occultations where the reflection occurs at an altitude that is inaccurately known, including over land, in the numerous cases over ice in the polar regions, and is also the case when the reflection takes place not at the surface but at some elevated refractivity gradient. Further work will be required to allow the interpretation of all these cases.

Also, since this procedure is interferometric, and the reflected signal is of smaller power than the direct, thus more limited in Signal to Noise ratio (SNR), the use of higher sensitivity instruments, and of wider bandwidth, should in general be expected to be beneficial. Notably, the future COSMIC-2 mission should provide the possibility to track these reflections with higher SNR. On the other hand, the COSMIC-2 mission is focused on tropical latitudes, where occultations present less frequently a clear reflection. However, the distribution shown in this study also indicates that in the higher latitudes seen by COSMIC-2, in both hemispheres, there is already a significant rise in the number and clarity of reflection events above the tropical minimum. It is to be expected that further studies with COSMIC-2 data will provide further insight on the information content of reflections.

*Data availability.* The ensemble of occultations processed, with their attributed SVM flags, is available freely from www.romsaf.org as an experimental product (registration required).

*Competing interests.* The authors do not have competing interests concerning this work.

*Acknowledgements.* The authors are grateful to anonymous reviewers for their comments, and to the following providers of GPSRO data for
5  offering access to their data products: the University Corporation for Atmospheric Research (UCAR), and the National Space Organization of Taiwan (NSPO) for COSMIC data.

This study received financial support from the Canadian Space Agency (CSA), from the European Organisation for the Exploitation of Meteorological Satellites (EUMETSAT) through its Radio-Occultation Meteorology Satellite Application Facility (ROM SAF), and a contribution from the Spanish grant ESP2015-70014-C2-2-R. Some of these grants are partially funded by the European ERDF/FEDER
10 Fund. The authors are members of the ROM SAF, which is a decentralized EUMETSAT facility.

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

**Table 1.** Comparison between dedicated GNSS reflectometry missions, and reflected signals during GPSRO radio occultation events. Example past or planned missions are shown, with one or the other technology. Launch years are included. This paper is solely adressed to reflections during GPSRO.

|  | Dedicated GNSSR | Reflections in GPSRO |
|---|---|---|
| **Geometry:** | near nadir ($\sim 45° - 90°$ elevation) | grazing propagation, nearly tangential ($\sim 1°$ elevation) |
| **Scattering:** | mostly diffuse | only coherent component acquired |
| **Existing, past, or planned:** | TDS-1 (2014), CYGNSS (2016), GEROS-ISS | e.g. CHAMP (2000), COSMIC (2006), METOP (2006), etc |
| **Peer-reviewed bibliography:** | 100+ articles | $<10$ |

**Table 2.** Percentage of reflected signals within different latitudinal belts, and reflecting surface typology. Reflection is here defined as SVM output value $> 0.25$.

| Latitudes [deg] | ALL | OCEAN | LAND |
|:---:|:---:|:---:|:---:|
| $|lat| \leq 20$ | 18 | 22 | 6 |
| $20 < |lat| \leq 50$ | 35 | 45 | 10 |
| $50 < |lat| \leq 70$ | 50 | 69 | 13 |
| $70 < |lat| \leq 90$ | 60 | 68 | 52 |

**Table 3.** Cross-correlation between the percentage of reflected signals in COSMIC ocean events, and other atmospheric variables. Correlations are based on monthly averaged values over $10° \times 10°$ ocean cells (see an example in Figure 8), except the optical heterogeneity, which is a correlation of individual events. All the variables are obtained from ECMWF ERA Interim analysis, except those obtained from the occultation profiles (first two rows), and the heterogeneity, from Environment Canada's global forecasts (last row). ERA Interim variables that depend on height have been averaged over the lowest 10 km. $\Gamma$ is the temperature lapse rate, and $\Gamma_s$ is the Saturated Adiabatic Lapse Rate.

| Variable: | Cross-correlation |
|---|---|
| Min. gradient of refractivity | 0.63 |
| Max. bending angle | -0.58 |
| Sea surface temperature | -0.80 |
| Column of water vapor | -0.78 |
| Vertical integral of total energy | -0.77 |
| Atmospheric temperature | -0.75 |
| Water vapor pressure | -0.69 |
| Sea wave height | 0.04 |
| Relative Humidity | 0.26 |
| Wind at the surface | 0.43 |
| $\Gamma - \Gamma_s$ (defined positive) | 0.64 |
| Optical Heterogeneity | -0.64 |

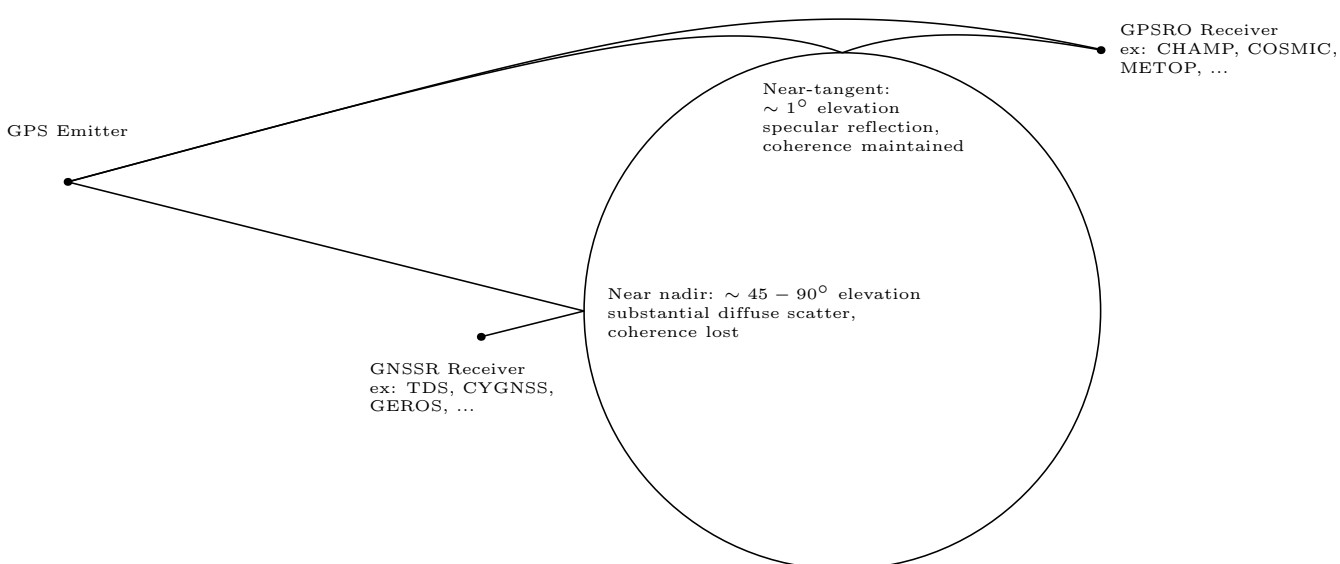

**Figure 1.** Diagram showing the concept of atmospheric occultation, together with the grazing reflection during the occultation, which is the focus of this study (upper part of the diagram). For comparison, the concept of GNSS reflections (GNSSR), at higher elevation angle, is shown on the left. At very low elevation angle, the coherence of the emitted signal is often not lost, even if the signal bounces off a surface that is not perfectly smooth (for instance ocean waves, see main text). During a reflection at high elevation angle, which is the concept behind several recent missions (TDS, CYGNSS, GEROS,...), normal ocean waves destroy coherence.

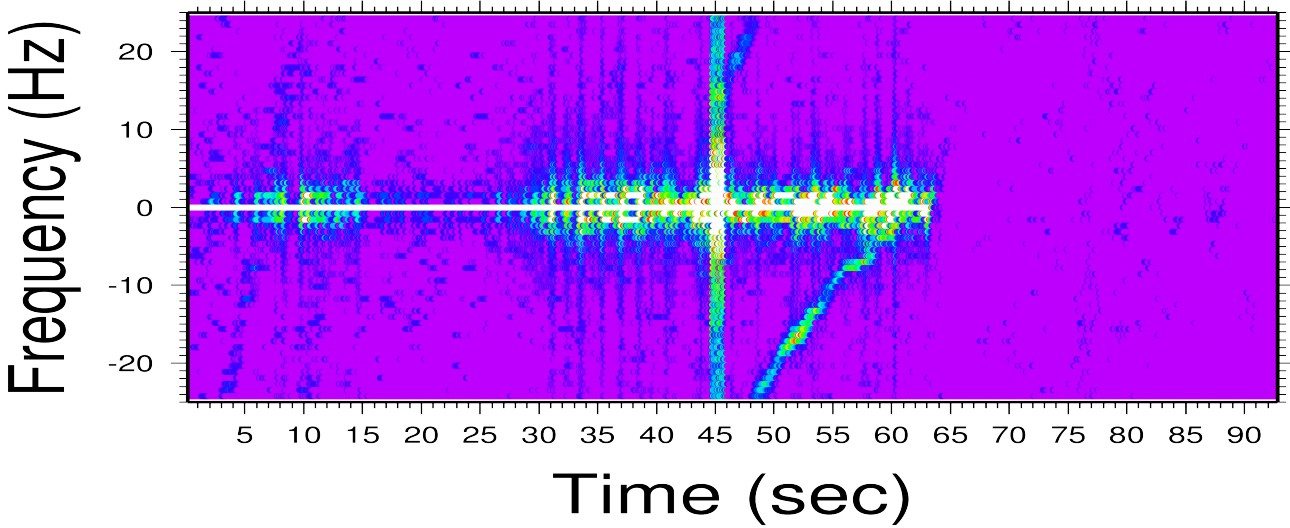

**Figure 2.** Example radiohologram of setting COSMIC occultation C001.2011.219.10.43.G15, which presents a reflection. The main signal has been nearly stopped by beating it against a filtered reference. Both components have a narrow spectral width, but the supplementary tone also shows aliasing (at 46-48 s, +20 Hz). Both tones finally merge. This final section, where the low troposphere is being scanned, shows spectral broadening in both the direct and reflected signals.

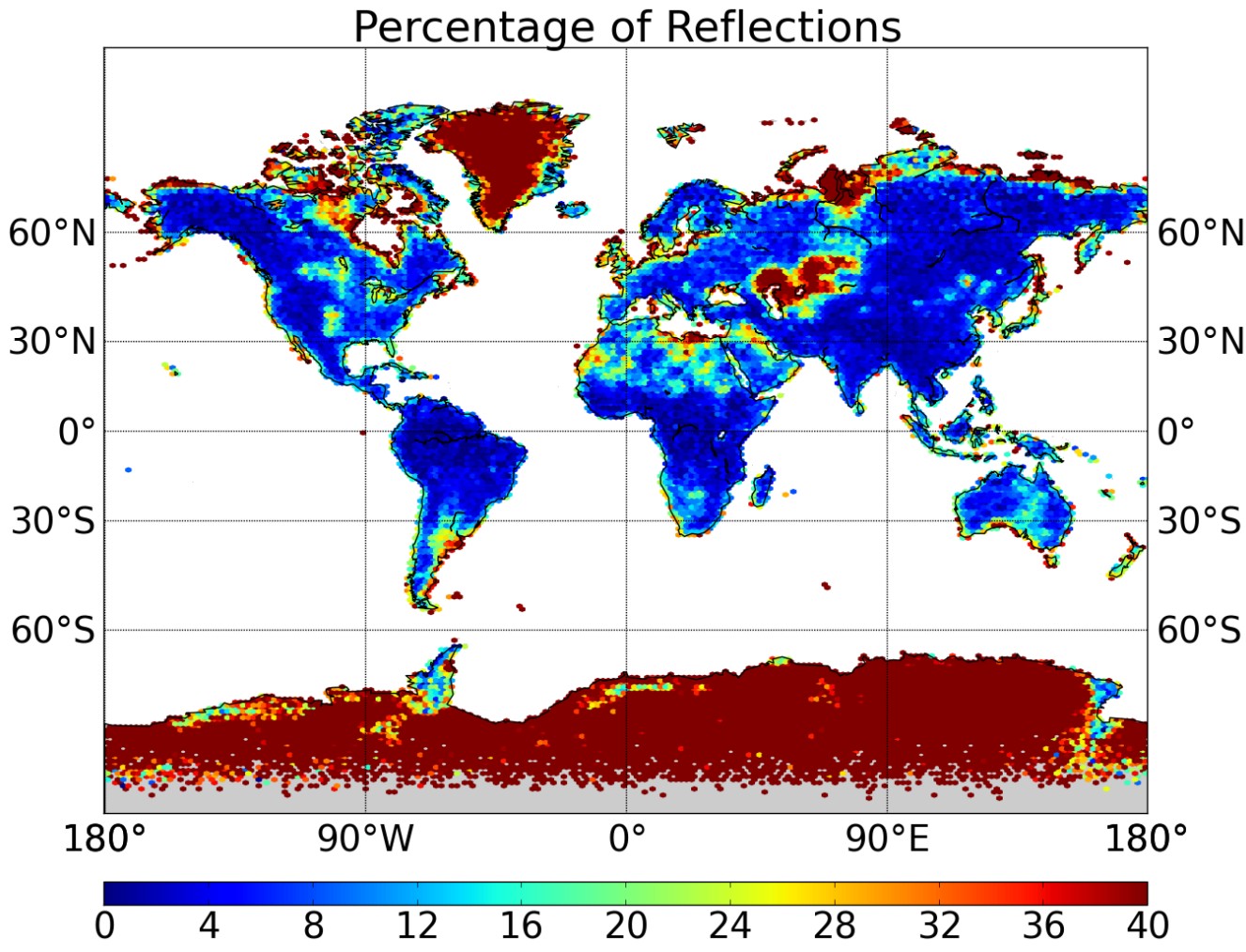

**Figure 3.** Map of percentage of GPSRO events identified as showing a reflection, over land. The map was compiled with ∼1,320,000 land events. Reflection events, here identified as those presenting SVM>0.25, mostly occur over smooth terrain, free of thick vegetation (deserts, tundra, grasslands), and over continental ice. The largest percentages are found over the ice sheets. The color scale is saturated at 40%. See also Supplementary Movie 1, which shows the seasonal cycle of the median SVM output value over land.

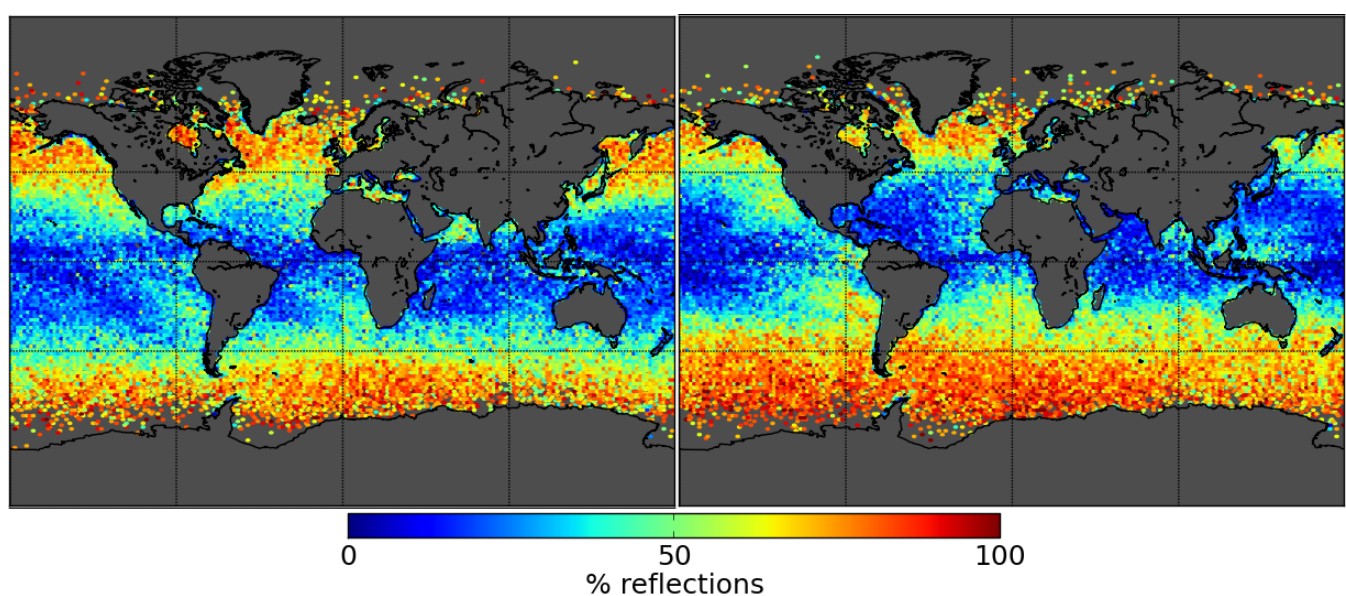

**Figure 4.** Statistics of GPSRO events identified as showing a reflection, over the ocean, obtained with the COSMIC constellation from 2006 to 2015. Percentage is shown during January and February (left), and during July and August (right). Note that the color scale is different from Figure 3. When the number of events per pixel is too small to derive sensible statistics, these are not shown. In polar oceans, reflections are also very frequent (see also Figure 7). See also Supplementary Movie 2, which shows the seasonal cycle of the median SVM output value over ocean.

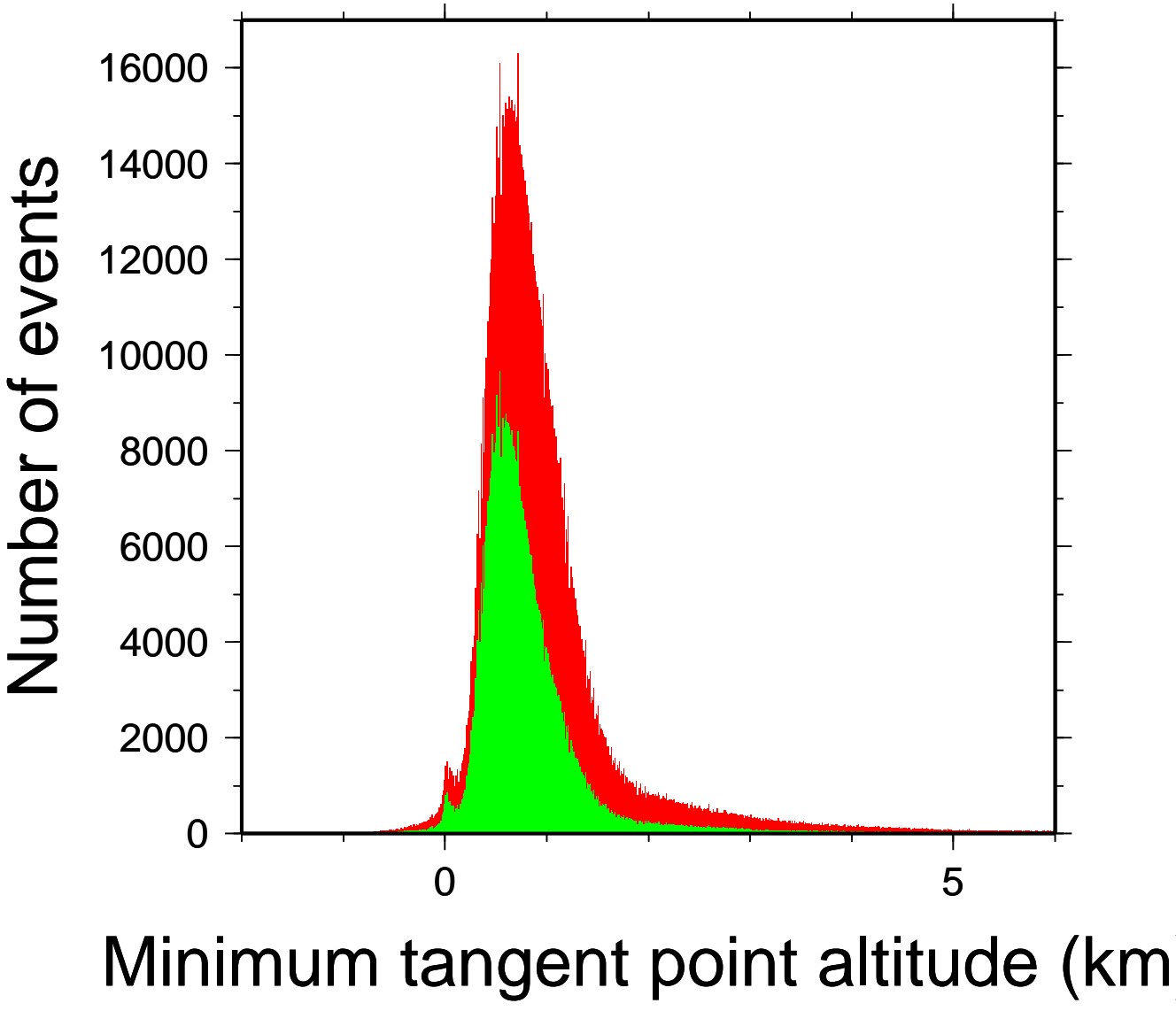

**Figure 5.** Histogram of lowest altitude profiled in ocean events that have been flagged as likely reflection (SVM>0.25, in red) and very likely reflection (SVM>1, in green).

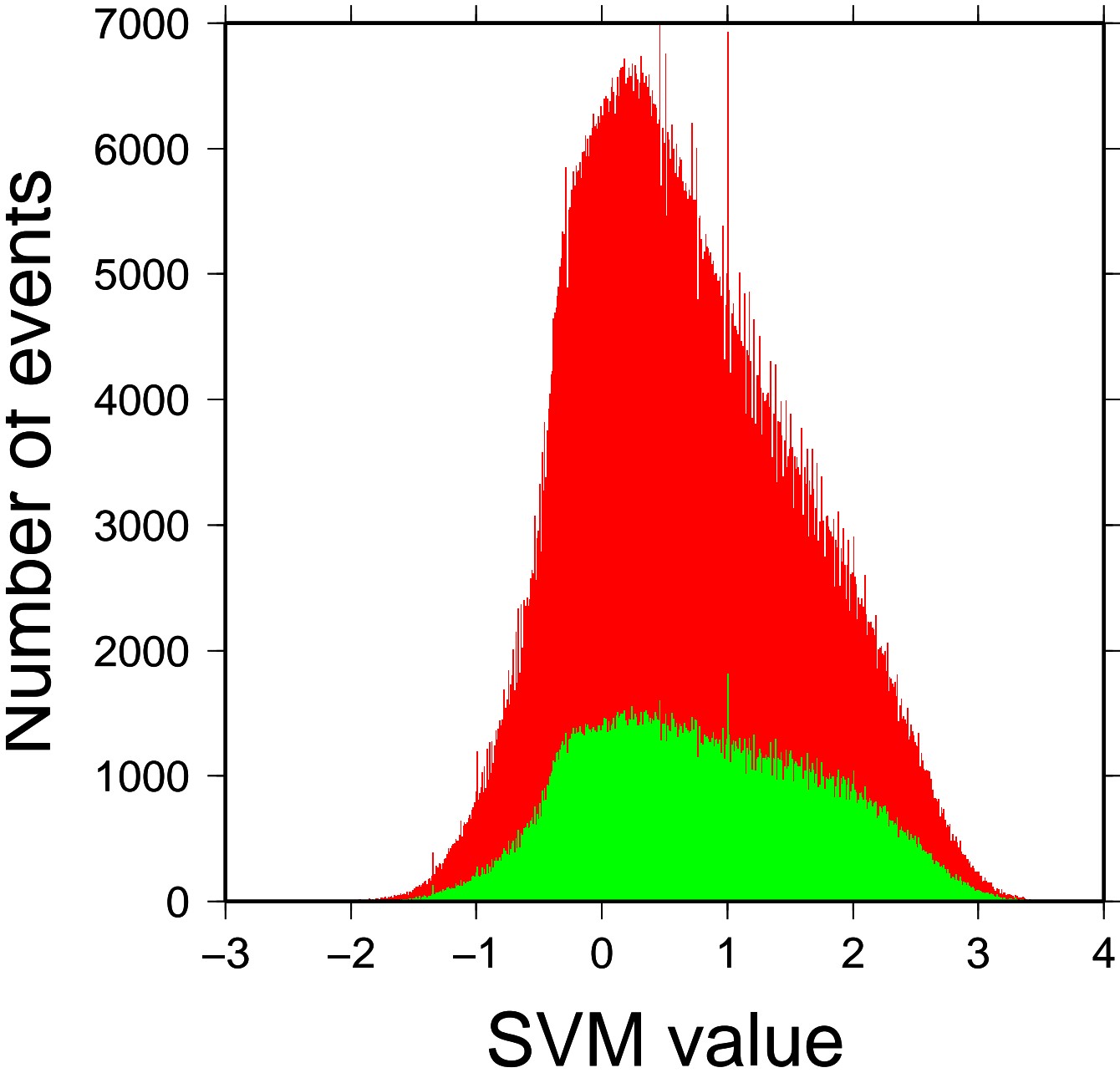

**Figure 6.** Histogram of SVM values for ocean events that reach the lowest kilometer of the troposphere ($H_{min} < 1km$, in red) and the lowest 500 meters ($H_{min} < 0.5km$, in green).

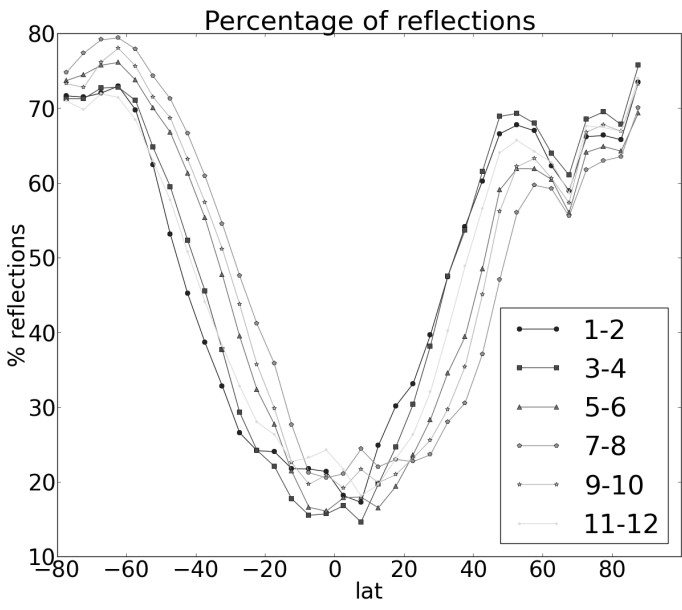

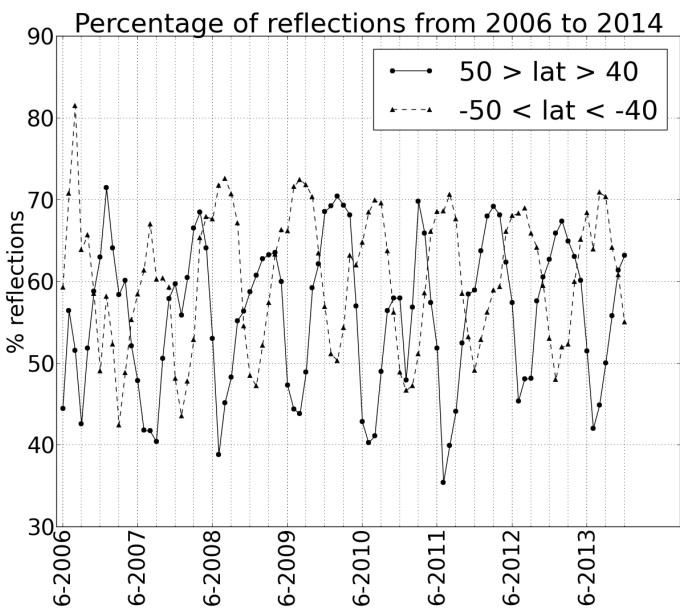

**Figure 7.** Statistics of reflection occurrence in events over ocean, obtained with the COSMIC constellation from 2006 to 2014. Top: percentage reflections as a function of the latitude (degrees), binned by bands of $5°$, over all 6 bi-monthly groups in the year. The curve shows a shift northwards during boreal summer, southwards during austral summer. Bottom: monthly time series of percentage reflections in the northern and southern midlatitude belts, between $40°$ and $50°$, showing the seasonal variation, and its opposite phase in each hemisphere.

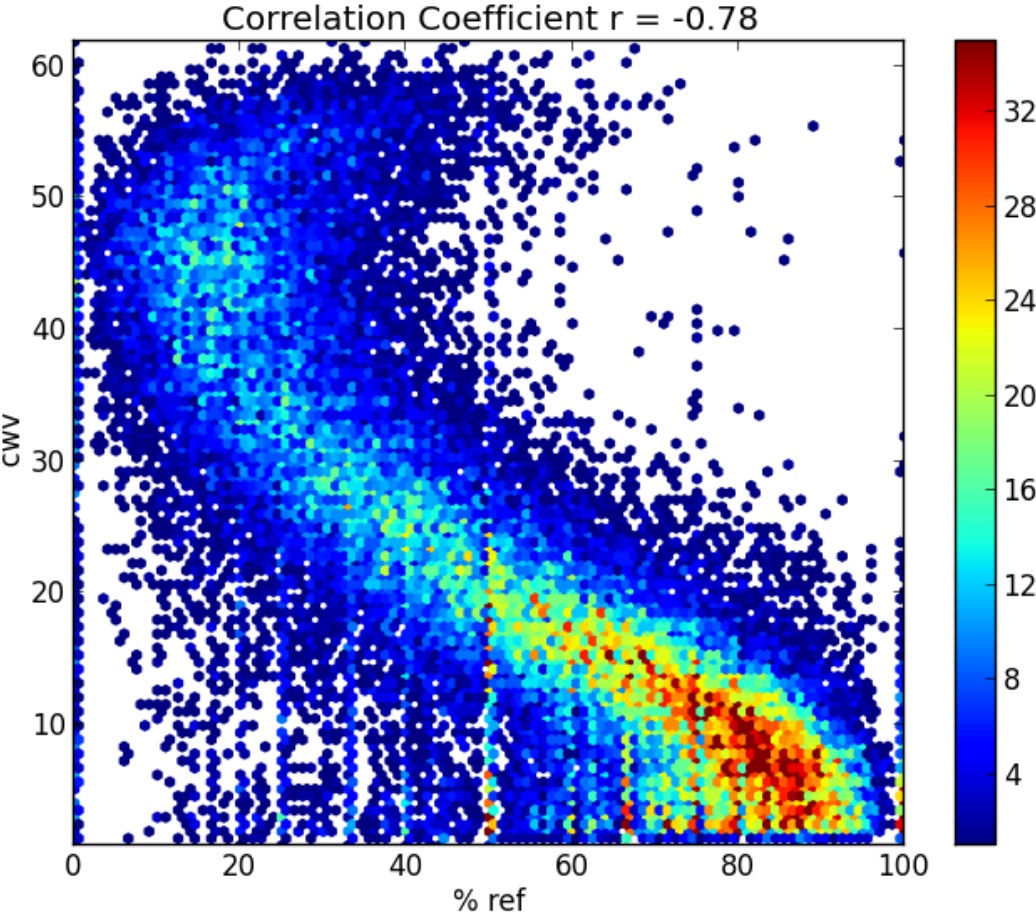

**Figure 8.** Two-dimensional histogram of occurrence of reflections, computed monthly on $10° \times 10°$ ocean cells, against the column of water vapor averaged over the same cell and month. ECMWF ERA Interim data have been used to estimate the values of the Column of Water Vapor (CWV, in $kg/m^2$). The color scale represents the number of (cell x month) counts, and it has been saturated to 35 for clarity. The resulting cross-correlation coefficient between reflection and CWV is $r = -0.78$.

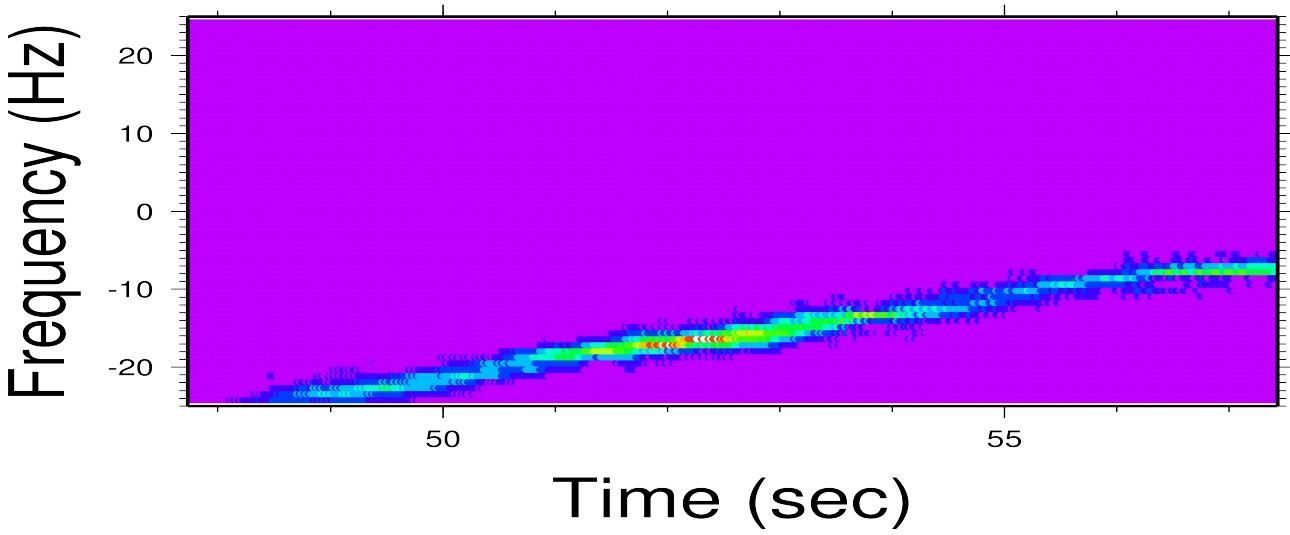

**Figure 9.** The location in time of the reflected signal is determined cross-correlating the radio-hologram and a template image of the typical reflection signature. In Figure 2, this identifies as the interest interval the time section shown here. Only this interval is further analyzed for the purpose of reflections. All frequency components above $-5$ Hz are set to noise level, which rejects the direct signal, which was close to 0 frequency. After inverse Fourier Transform of the filtered hologram, a phase lock loop (PLL) can track the signal identified as the reflection, obtaining $f_R(t)$.

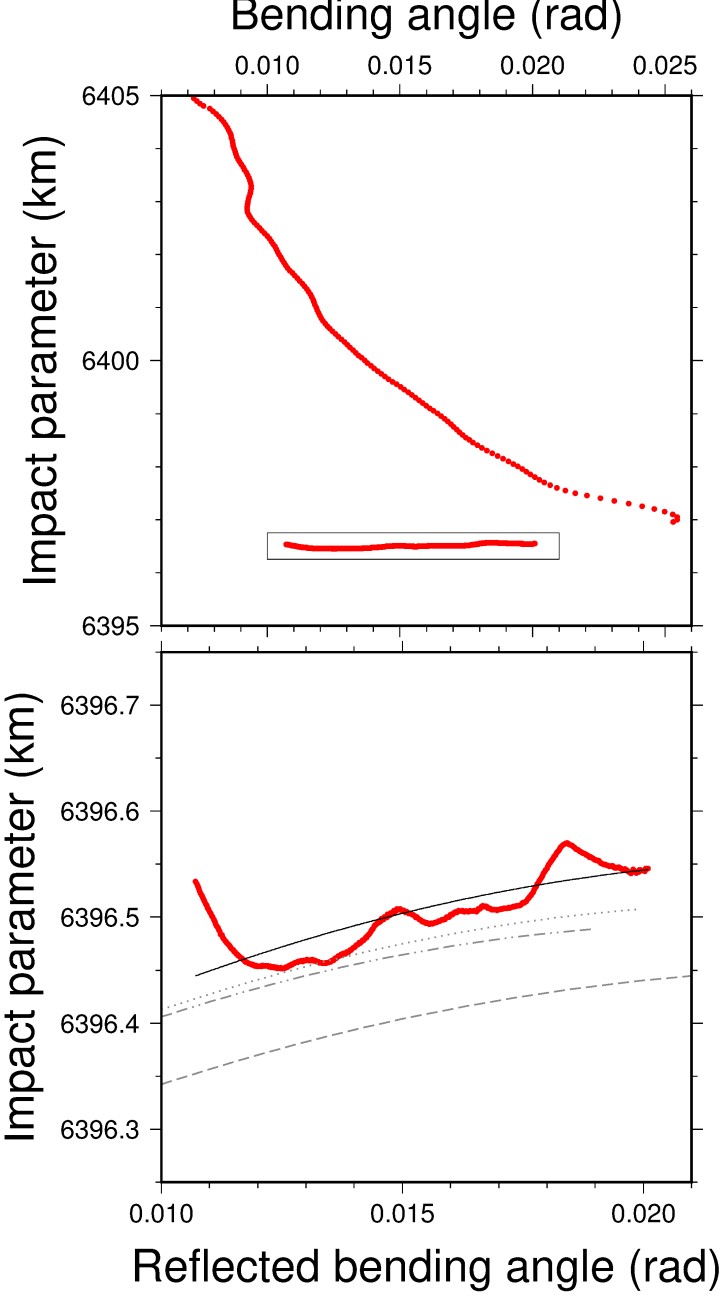

**Figure 10.** In the upper panel, profile of extracted bending and impact parameter, including the standard direct branch, as well as the reflected, from the same occultation in Figure 2. The box around the reflected branch is zoomed in the lower panel. The reflection was extracted as detailed in Section 4 (thick red). A parabolic fit to it is shown in solid black. The COSMIC Data Analysis and Archival Center (CDAAC) also offers profiles of refractivity $N(h)$, which allow the evaluation of the expected $\alpha_R$ and $a$ using Equation (3). These are shown in grey, using refractivity obtained from the direct data (atmPrf file, dashed-dotted), from a background NWP (ecmPrf file, dashed), and from a 1DVar estimation (wetPrf file, dotted).

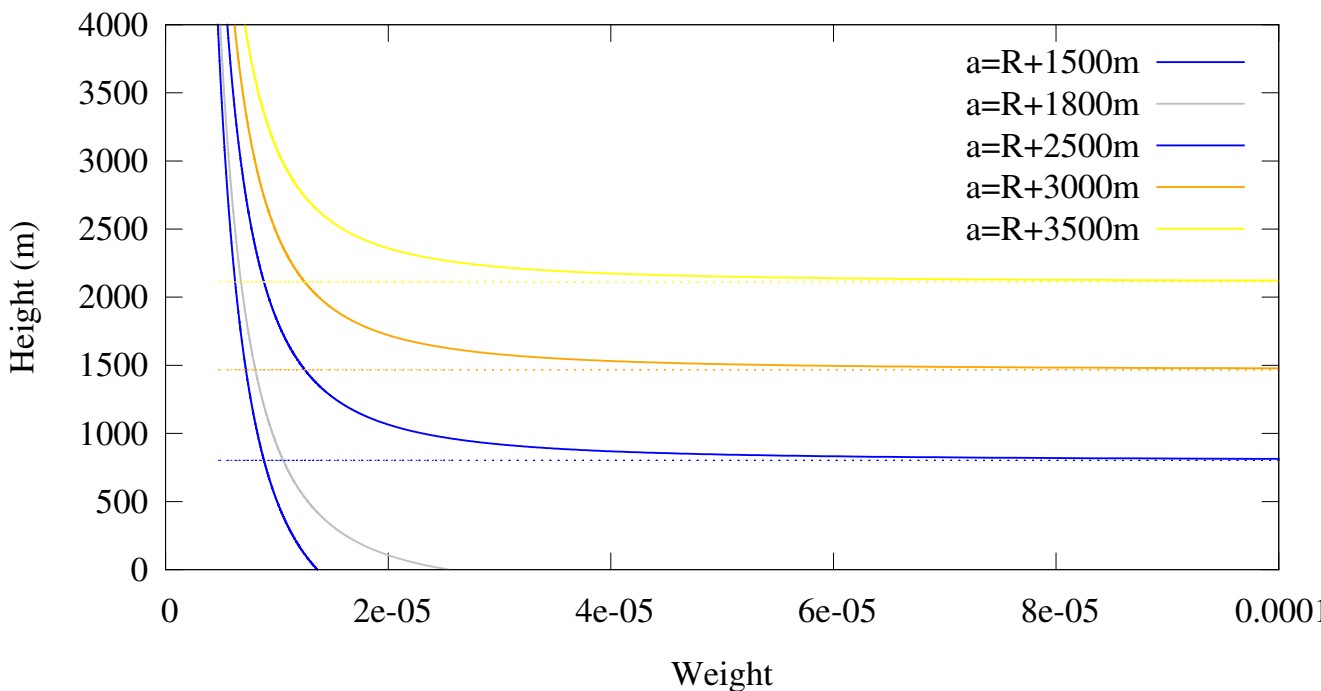

**Figure 11.** Kernel weight (adimensional) dependence of several paths, with respect to the local refractivity properties of the profile. Weights for three direct paths are shown, with impact parameters ($a = R + 2500m, R + 3000m, R + 3500m$). These kernels present an asymptote (shown dotted, an integrable singularity), and are zero below the asymptote. Given the concentrated weight near the respective asymptote, the information provided by each observation is very independent from each other, leading to an ensemble that provides information of high vertical resolution. Weights for two reflected paths are also shown, with smaller impact parameters ($a = R+1500m, R+1800m$). These do not show any narrow concentration of weight. They provide information that is different between them, but much less independent vertically. With the refractivity profile used, the apparent horizon was approximately located at $a_S = R + 1900m$.

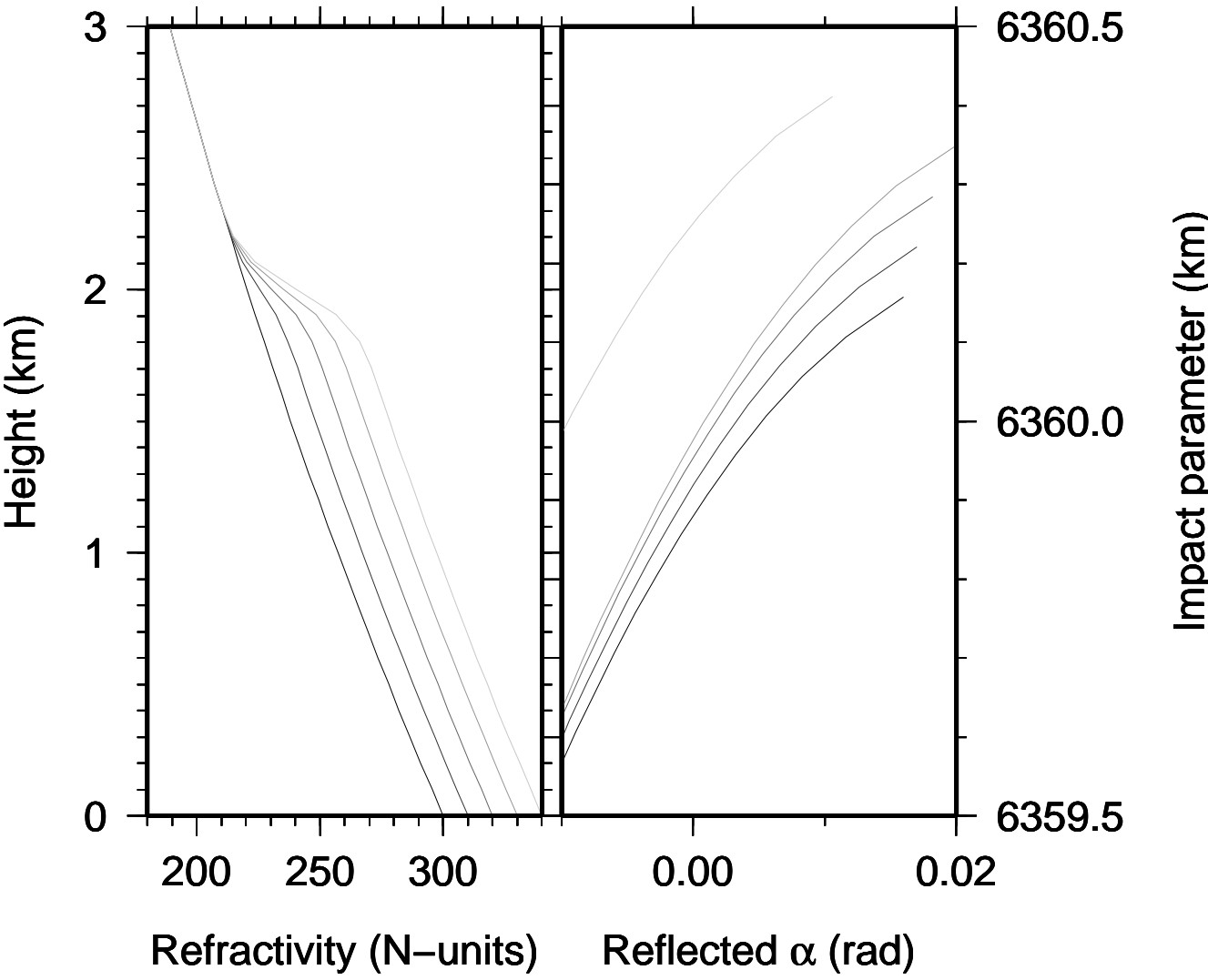

**Figure 12.** Synthetic case to demonstrate sensitivity of reflection under ducting. The left panel shows a family of synthetic atmospheric refractivity profiles, generated as an exponential function, with a more refractive lower troposphere. The transition is modeled as an error-function. The transition to this extra refraction is very steep, which is common in superrefractive or ducting layers. The right panel shows the corresponding series of reflected bending and impact parameters (see the lower pannel of Figure 10 for a case with real data), as evaluated by Equation (3), and indicates that the profile of reflected bending is sensitive to the refractivity step across the transition.