# Peer review of "Information content in reflected signals during GPS Radio Occultation observations"

_Atmospheric Measurement Techniques, 2017_

## Referee Comment (RC1) · Anonymous Referee #1 · 11 Oct 2017

The paper presents an overview of the impact and potential of using reflections in Radio Occultation observations; from their detectability to their "possible" use in the retrieval of bending angles. It is nicely organized, complete, and the topic is explained thoroughly, including correlations to several other parameters. It is appreciated to finally have a "reference" paper describing the state-of-the-art of reflection in RO. The paper is clearly written for a wider audience than just the RO community, since it provides also basics of the radio occultation theory. These basics are however nicely in context to improve understanding the extension of this theory (which is normally applied to retrieve bending angles from the direct rays) to reflected rays. Future work on this, where the information in the reflected ray is further investigated, is encouraged.

All in all, this paper should be published after a minor revision. Some specific com-

ments to improve the clarity of the manuscript are the following:

- Several times reference is made to the coherence of the "reflected" signal. Could you please add further information in the introduction in order to better address this concept (the first reference to the coherency is made at Page 2, row 30 and then again at page 3, row 18)? It would also be nice to explain why the receiver might lose tracking in case of lost coherency (assuming phase coherency... which is basically a phase which is varying randomly, because of complex scattering happening around the reflection point). Finally, it would be nice to also explain why a RO receiver can potentially track reflected signals only at very low grazing incidence (because of polarization, see first comment in the technical corrections below, but likely also because of the "phase/range" difference between direct and reflected paths that does not allow the PLL/DLL to be locked on both - but this is valid for standard closed loop and open loop using only a Doppler model like the one adopted on GRAS. It would be helpful in that context to clarify whether the COSMIC receiver has a full code/phase open loop tracking).

- Page 2, row 30: here the concept of "interferometric" phase is introduced. This is a crucial step of the processing - it is worth to provide an explanation of what it is and why it is used.

- Page 10, row 19-21: super refraction is, in general, not ducting or, in other words, ducting is the upper boundary of super refraction and is normally defined as "anomalous propagation". The electromagnetic propagation phenomenology is different between the two. In case of super refraction (when the refractivity gradient is between $\sim$-87 and -157 km$^{-1}$) you still have "rays" bending towards the Earth's surface, with a bending angle which is increasing with the magnitude of the gradient (associated with a "signal" power which is decreasing with the square of the distance). In case of ducting (refractivity gradient < -157 km$^{-1}$) rays are trapped like in a waveguide, with very small decrease of power while they propagating. Rays may or may not enter the duct, depending on the incidence angle, and if they enter, may exit very far away from

the point where they entered.

- In the manuscript it is unclear when it is referring to super refractivity or to ducting. Depending on the refractivity gradient, on the ducting length, on the geometry (rays are in any case grazing the ducting layer, so they can theoretically enter the duct), rays might not be reflected from the surface closer to the expected specular point (if they are entering the ducting they will propagate along the duct before exiting). I guess that

1) trying to compute the correlation between maps of reflections and duct maps would be a good exercise and maybe can show some interesting features

2) moreover, could you please add in Figure 12 (as a legend or in the caption) the refractivity gradient magnitude and also add a third panel with the actual direct bending angles?

In this "family" you likely have both cases of super refraction and ducting (in particular the profile shaped as error function. Is what you wrote at Page 15, row 20 is true? First, propagation is also defined by the refractivity gradient, not by the surface refractivity value. Moreover, being a simulation (or application of forward model, which does not model at all the propagation within a duct), the "ray" maybe cross the duct because the incident angle (the angle with the normal to the duct) is small enough and it is "refracted" into the duct. As a first thought, in this case one might expect bigger bending angles for the same impact parameter (or smaller impact parameters for the same bending). But, looking at the family of bending angle profiles in Figure 12, right panel, this is showing exactly the opposite. The second thought is one the incidence angle, which is decreasing with increasing N gradient (more bending towards the surface), thus the overall bending angle defined by equation 3 is becoming smaller and smaller because the second term in this equation is increasing. Is it correct? Could you please add a sentence on this?

- Page 12, row 13: if taking the fR from the interferogram shown for example in Figure 9, fD is always zero and thus fR = fI? If this is true, where is the fD taken from? From

the measurements on the direct link (thus from the original carrier phase of the signal)? Taking simply the time derivative of the carrier phase? But it is well known that, in the lower troposphere and in particular in the sub- and tropical regions, atmospheric multipath often occurs. This means that the instantaneous (excess) Doppler components cannot be estimated by the carrier phase time derivative. Wave optics techniques (FSI for example) should then be applied. Thus is the fD(t) for a particular time t when the fI(t) of the reflection is also present, computed with a wave optics approach? Could you please clarify this in the paper?

- Page 16, row 20: Somehow doubting that the reflected beam will allow extracting information if the direct one is lost. This would be a "nice" case, but in reality the receiver will encounter low SNR in the lowest part of the atmosphere, is unable to track the direct beam, so why should the reflected one (which encountered even more of that lowest, variable part of the atmosphere) still be present?

- General: it might be worthwhile to also think about corrections/removal that can be applied based on knowledge of the reflected beam. This could be very valuable for all the situations where the direct beam is already very noisy (and no reflected one is easily identifiable). Can you include a few words on this option?

- General: would be nice to include some general statement on what to expect from e.g. COSMIC-2 with its high gain occultation antennas.

"Technical corrections"

Page 2, row 22: "either by polarization". This is not strictly true. The signal is right hand circularly (RHCP) polarized. The receiver antenna onboard the LEO must be sensitive to the RHCP polarization only, with a very low sensitivity to the cross-pol component (the left hand one). GPS receiver cannot separate the two components at all. The remaining energy in the LH component is something like a noise. In case of reflection with low incidence angles (∼nadir reflections, typical for a GNSS-R receiver), the polarization of the reflected signal will be LH. Thus to make the receiver "sensible" to

such signals, the antenna should be a LHCP one. Again the receiver is not separating anything.

Page 2, row 26: what does it mean that the spectral distribution is compact? Maybe that the spectral components are within a small bandwidth? Use another adjective than compact...

Page 3, row 3: "signals" instead of "profiles"

Page 3, row 4: "observations to be inverted to estimate the profile of..." instead of "measurements of the profile..."

Page 4, row 2: it is certainly worth to cite ESA's plans on this, but maybe investigating is better than planning? And there is an Eart instead of Earth in previous line.

Page 4, row 6: grazing "observations" instead of grazing "angles"

Page 4, row 28: "by a PRN code" instead of "by the PRN".

Page 4, row 33: what does it mean "orbital Doppler"? Is the Doppler experienced by the signal due to the relative radial movement between GNSS and LEO considering vacuum propagations? Please specify.

Page 5, row 1-2: what are you referring with "electromagnetic noise"?

Page 5, row 3: again, is the Doppler subtraction referred to the "orbital Doppler" not better defined at Page 4, row 33? please clarify.

Page 5, row 4: what do you refer with "frequency broadening"? please clarify.

Page 5, row 12: what about L2C tracking, which might be better than L1C/A?

Page 5, row 20-21: this is the standard frequency beating phenomenon. Such details can be avoided.

Page 6, row 22: you are naming the threshold you need to define simply as SVM (also later on in the paper) Could you please find another acronym/name to such threshold? SVM refers to "Support Vector Machine" which is a method, not a numerical value/threshold (see for example row 27 same page).

Page 7, row 22: reference to a supplementary movie. If you think it is ok, discard this comment. People will mostly download the paper and simply read it. Is it potentially worth to put the movie somewhere in an open repository (e.g., DropBox, Google cloud, ... ) and make reference to the repository? See also Page 8, row 29

Page 8, row 10: I think that Ulaby could also be referenced well before. Probably in discussing coherency, or simply when you talking about electromagnetic reflection.

Page 8, row 28: Looking at Figure 7, I'd say that the fraction of reflections at lat > 45deg is always large, but between 45% and 65%. Unfortunately the figure uses grey levels, so it is not very clear. I suggest to use also different line styles, in order to better show the seasonal similitude (if I'm not wrong, in summer/winter there are always more/less reflections in both hemispheres). Maybe it is worth to point it in the paper also discussing figure 7.

Page 8, row 35: "opposite" instead of "contrary"?

Page 11, row 33: Could you better define the 2D template used to cross-correlate the radio-holographic image?

Page 12, row 5-9: Could you please specify as a further detail for a more generic audience why the reflected signal is always going from -25 Hz to 0 in an interferogram like the one shown in Figure 2? Or at least provide a reference for this (I guess that there is a theoretical explanation here : Beyerle, G., K. Hocke, J. Wickert, T. Schmidt, C. Marquardt, and C. Reigber, GPS radio occultations with CHAMP: A radio holographic analysis of GPS signal propagation in the troposphere and surface reflections, J. Geophys. Res., 107(D24), 4802, doi:10.1029/2001JD001402, 2002.)

Page 12, row 19-21: this is unclear. What is this formal precision of the inversion? And where is the evidence that the "thickness" of the "line" is larger than the precision? Is

this because no noise is visible? This is also repeated at page 14, row 10-11.

Page 12, row 25-27: unclear. Are you inferring that atmospheric multipath is neglected (or rays are disentangled after some preprocessing)? Why are you talking about "diffraction" here? Diffraction from what? Earth's surface?

Page 13, row 10: smaller typo

Page 13; row 18: "by twice the incident angle" instead of "... grazing angle".

Page 12, row 29-31: I guess that the wording "local spherically distributed" refracted field can be used instead of its explanation.

Page 16, row 21: If you have a loss of track, you are loosing both the information carried by the direct and the reflected signal. Instead if I well understood, in the time series of observations, the reflected contributions to the bending angle characterizing the lower impact parameters come before the direct contribution. In this case it makes sense that in case the signal is lost and, if any, reflections are "tracked" they can be used to eventually fill in the gap. Not sure but if you have pure open loop data, you can always reconstruct both (if SNR allows it). Could you please better address this?

Page 16, row 24-27. Not clear. Please, rephrase it.

Table 2, caption: "surfaces typologies". Again the reference to SVM value (see comment to Page 6, row 22)

Figure 1, caption: GNSSR at higher elevation (strictly speaking in this case the elevation angle is smaller...)? What does it mean? I'd use nadir looking. Also, here you use GNSSR, while otherwise you use GNSS-R.

Figure 2, caption: "setting" instead of "descending". What does it mean "winding"?

Figure 3: with the uncertainty of the tangent point being a few km, are land reflections over small islands truly land ones, or could also be sea ones? Maybe add a pre-caution.

Figure 3, 4: it seems the Arctic is neither land nor sea. Are there no reflections observed there?

Figure 4, caption. You are referring to COSMIC here. Not sure in the text this is made. And, for instance, it is not made in Fig 3, caption, but it is made in the caption of Figure 7. Again also here is made reference to the "supplementary movie.."

Figure 8: at 50% ref, there seems to be a vertical line for all cwv, is this some threshold trigger?

Figure 10: might be better in impact height (IP - RoC)

Figure general: they need to be made more coherent, font sizes are sometimes very large, sometimes okay, sometimes small. Units missing, etc.

---

## Referee Comment (RC2) · Anonymous Referee #2 · 17 Oct 2017

The paper presents an overview of the detection of reflected signals in radio occultation events and discusses the potential use of additional information that could be derived from such data. It is well readable, also covers some basics and appears complete, although the authors could sometimes be more concise. Some statements and claims are made that should be clarified or corrected, see my comments below.

To summarize, the paper is acceptable for publication after addressing the following issues.

- Page 2, line 20: "Also, at this very low elevation, and unlike at higher incidence angles, a reflection does not lead to a reversal of the polarization."

I think the description of reflection off an infinite surface by use of Maxwell's equations

does not entail a dependence of the polarization from the incidence angle as claimed above. The authors need to either prove this claim for the geometry at hand or give some reference.

- Page 4, line 28: "GPS satellites carry atomic clocks, which produce pure sinusoidal tones. Before emission, these tones are modulated by characteristic bitstreams. ..."

Unless I misunderstand things, the GPS carrier signal is primarily generated by some (highly stable) oscillators for the Lx frequencies, whose signal is modulated with information derived from the atomic clocks, plus ancillary data. There exist different technologies for atomic clocks (e.g. Cs or Rb based), whose details are of little interest as far as the emission, reception, and further processing of the carrier signal is concerned. Since this is also probably not relevant for this work, I recommend that the authors correct or remove this claim there and in other places of the manuscript.

- Page 6, line 27: "... radio occultation events with SVM value greater than 0.25 are identified as reflections. "

The use of the term "SVM value" is probably some slang. A couple lines above the term "scalar" was used. Further down, "... SVM threshold." I would recommend to use a more coherent term for the output variable of the SVM.

- Page 8, line 35 to page 9, line 3: "These patterns, consistently associated with geographic and seasonal features, do not suggest a direct relationship with instrumental problems or performance. They may, at most, be linked to instrumental performance issues if these arise under certain geophysical conditions related to seasons or geography."

I am not sure what the authors wanted to tell here. Is there evidence for variations of instrumental performance, for fundamental instrumental limitations, or other? Please rephrase.

- Page 9, line 7: "... The fields from the ECMWF ERA Interim analysis are used here

to determine the weather state. The correlations presented in the following are all computed using monthly averages evaluated on cells of 10°x10° over the oceans, with the reflection fraction compared against several environmental variables."

How useful is the use of monthly averages for determining the correlation of the fraction of reflections with several atmospheric variables, or quantities derived from the weather situation? For "slow" variables such as SST this is certainly fine. But given the large cross-correlation for some of the quantities given in table 3, one might be asking if using the actual situation instead of some average might lead to different results. Did the authors investigate this?

- Page 9, line 32: "... must not be directly linked ..."

Should "must not" rather read "cannot"?

- Page 10, line 6: "As mentioned above, the significant sea wave height is nearly un-correlated (r=0.04), although wind speed over sea has a moderate positive correlation (0.43). This was somewhat unexpected, as stronger winds correspond to rougher sea surfaces, which could seem to link to less chances of coherent reflections."

Not sure if it is relevant, but for measurements of wind speed over sea (e.g. scat-terometry and altimetry), instead of SWH the correlation with smaller scale features of the sea surface such as capillary waves is exploited. These are important to the description of reflection and damping of (Ku/C band) signals at larger incidence angle.

- Page 11, line 15: "that the reflection flag, either a qualitative present/absent, or the quantitative SVM value, stems from the observation, and that the knowledge that an oc-cultation is expected a priori to show lower OMB difference than an average occultation is already a supplemental NWP value to the standard profiles of direct non-reflected signals."

Did anybody already see positive impact from using that information? The paper by Healy says that this appears to be difficult in practice. So why is it "already a supplemental NWP value"?

- Page 12, line 19: "The formal precision of the inversion ..."

Can the authors please explain what they mean by "formal precision"?

- Page 13, line 11: "This is a physically new phenomenon that must be included ..."

I don't think that reflection is a new phenomenon, it is just that it needs to be taken into account. Recommendation: discard "physically new".

- Page 13, line 18: "Due to reflection, the direction of the ray suddenly changes at the surface, ..."

"suddenly"? Use a less prosaic description of trivial reflection.

- Page 14, line 12: "Interestingly, the slope of the reflected profile is very sharp, when compared with the direct profile. ..."

When looking at eq.(3) and taking the derivative w.r.t. a, one has two contributions of different origin. The second term on the r.h.s is purely geometric and thus more or less trivial, while the first one involves a derivative of the kernel K of eq.(4) that would be quite interesting to see a discussion about.

- Page 14, line 17: "but the fact of having reached the surface, ..."

It is actually an assumption here that reflection is off the (Earth's) surface, and not some reflecting layer, isn't it?

- Page 14, line 24: "For direct, non-reflected propagation paths, this dependency of the properties of the bending with respect to the atmosphere, presents sharp peaks at the respective tangent altitudes."

The kernel K has an (integrable) singularity, not just a "sharp peak". The following discussion on that page and also fig.11 is based on the naive assumption of some "sharp peak" and needs to be corrected.

- Page 14, line 32: "The reflected kernels always include the entire atmosphere, and are always sensitive to the low troposphere. Among the direct paths, only a few are sensitive to the low troposphere, and some may be missing. The reflected kernels have therefore the ability to fill any section where direct data are not sufficiently sensitive."

Is this "filling of missing sections" just some hope, or has it actually been done or shown? If not, remove or weaken the claim in the last sentence. Given the discussion in that paragraph and the first one on page 15, I recommend that strongly.

- Page 15, section 4.3: "Value of reflected data under superrefraction"

For the sake of completeness of the paper - and also for understanding this section - it would be good if the authors briefly explained superrefraction and ducting.

- Page 15, line 17: "The additional refractivity is shaped as an error function, leading to a vertical gradient that may be strong. ..."

This needs to be explained better. From fig.12 I would expect that the vertical gradient is strongest near 2 km height, so the relation to surface refractivity is only through the property of the employed model (assumptions). The conclusions described there are not clear to me. What am I missing?

- Page 16, line 19: "Although the reflected section of the profile contains less independent information than the direct section, it contains a few unique capabilities. It can resolve voids in the direct profile, for instance in the event of loss of track."

Can one really resolve voids in the event of loss of track? Has this claim been demonstrated somewhere?

Spelling:

- Page 18, line 11: "Marcquardt" -> Marquardt

Running a spell checker over the entire text might find a couple typos.

---

## Author Comment (AC1) · 14 Nov 2017

The paper presents an overview of the impact and potential of using reflections in Radio Occultation observations; from their detectability to their "possible" use in the retrieval of bending angles. It is nicely organized, complete, and the topic is explained thoroughly, including correlations to several other parameters. It is appreciated to finally have a "reference" paper describing the state-of-the-art of reflection in RO. The paper is clearly written for a wider audience than just the RO community, since it provides also basics of the radio occultation theory. These basics are however nicely in context to improve understanding the extension of this theory (which is normally applied to retrieve bending angles from the direct rays) to reflected rays. Future work on this, where the information in the reflected ray is further investigated, is encouraged.

All in all, this paper should be published after a minor revision. Some specific comments to improve the clarity of the manuscript are the following:

- Several times reference is made to the coherence of the "reflected" signal. Could you please add further information in the introduction in order to better address this concept (the first reference to the coherency is made at Page 2, row 30 and then again at page 3, row 18)?

We have added some information, in the introduction. Coherence is indeed a key concept, as the prime measurement is phase/Doppler, and the entire GPSRO concept, both for the direct propagation and these low elevation reflections, is interferometric.

It would also be nice to explain why the receiver might lose tracking in case of lost coherency (assuming phase coherency... which is basically a phase which is varying randomly, because of complex scattering happening around the reflection point).

We add that. The measurement is extracted either by hardware PLL, or through some software process later. Beyond losing PLL track, which is quite common but retrievable in postprocessing, we specify that lost track may mean "inability to record the signal for postprocessing".

Finally, it would be nice to also explain why a RO receiver can potentially track reflected signals only at very low grazing incidence (because of polarization, see first comment in the technical corrections below, but likely also because of the "phase/range" difference between direct and reflected paths that does not allow the PLL/DLL to be locked on both - but this is valid for standard closed loop and open loop using only a Doppler model like the one adopted on GRAS. It would be helpful in that context to clarify whether the COSMIC receiver has a full code/phase open loop tracking).

We did develop those in Sec 3.2, and is the primary purpose of Figure 1. We try to clarify, and advance in the Introduction that at this geometry, and for this wavelength, the sea surface leads to specular reflection, not to diffuse scatter. Primarily, current GPSRO receivers are designed to track the direct signal, and they evaluate where to find the direct signal in Doppler and pseudorange. The reflected can be tracked by these receivers if it falls within the modest bandwidth that the receiver is dedicating to the direct signal (for COSMIC, 50 Hz in frequency, and 300m in pseudorange). This is the case at low elevation. At higher elevation, the signal is outside this bandwidth, both in frequency and in pseudorange. If the receiver

dedicated a channel to the reflected signal, it could also be tracked at much higher elevation. This tracking would be either in phase if the scattering were coherent, or only tracking the envelope of the signal is the scattering at higher elevation angles would become diffuse (as it mostly happens in near-nadir observations, such as in UK TDS-1 or CyGNSS missions).
Existing devices (delay/Doppler receivers, ex. CYGNSS) can dedicate channels to track at several delay and Doppler offsets, but this is not the case with existing RO receivers. Interestingly, though, the reflected signal is mostly sensitive to the atmosphere, but not to the sea state, at low elevation, and it is thus an atmospheric measurement. It is the opposite at higher elevation, and if this signal was tracked, it would be a sea surface measurement, which is the case with CYGNSS.

- Page 2, row 30: here the concept of "interferometric" phase is introduced. This is a crucial step of the processing - it is worth to provide an explanation of what it is and why it is used.

We expand on this concept, on the creation of a reference, and the comparison against that reference. However, this concept was already present in earlier work, for instance (Beyerle et al, 2002). We reference to it.

- Page 10, row 19-21: super refraction is, in general, not ducting or, in other words, ducting is the upper boundary of super refraction and is normally defined as "anomalous propagation". The electromagnetic propagation phenomenology is different between the two. In case of super refraction (when the refractivity gradient is between ~-87 and -157 km^-1) you still have "rays" bending towards the Earth's surface, with a bending angle which is increasing with the magnitude of the gradient (associated with a "signal" power which is decreasing with the square of the distance). In case of ducting (refractivity gradient < -157 km^-1) rays are trapped like in a waveguide, with very small decrease of power while they propagating. Rays may or may not enter the duct, depending on the incidence angle, and if they enter, may exit very far away from the point where they entered.

The use of both was indeed inaccurate. We were focusing on the generic situation of cases where interpretation of direct (non-reflected) signals is potentially difficult, which includes both. We have parsed the document, and have chosen either "super-refraction" or "ducting" as appropriate.

- In the manuscript it is unclear when it is referring to super refractivity or to ducting. Depending on the refractivity gradient, on the ducting length, on the geometry (rays are in any case grazing the ducting layer, so they can theoretically enter the duct), rays might not be reflected from the surface closer to the expected specular point (if they are entering the ducting they will propagate along the duct before exiting). I guess that
1) trying to compute the correlation between maps of reflections and duct maps would be a good exercise and maybe can show some interesting features
2) moreover, could you please add in Figure 12 (as a legend or in the caption) the refractivity gradient magnitude and also add a third panel with the actual direct bending angles?

We reference to maps by (von Engeln and Teixeira, 2004), to note some coincident geographic anomalies. However we have no reason to think that there is a causal relationship between superrefraction/ducting and a reflection signal. The "reflected" signals are compatible (by the Doppler offset) with reflections at the Earth's surface, not with an elevated duct. We reword to indicate that we mention super-refraction and ducting for the following reasons
    1)  These are situations where reflection is more frequent than average.
    2)  These are situations where reflection is more useful (add new information to a challenging case for regular GPSRO).
We do add that the reflecting surface may in some few cases not be the ground or sea level. We consider this an issue of quality check.

In this "family" you likely have both cases of super refraction and ducting (in particular the profile shaped as error function. Is what you wrote at Page 15, row 20 is true?

Yes. We mean refractivity. See also the next comment. We try to clarify this in the text, indicating that the reflection is sensitive to the net change in refractivity across the ducting layer, a sensitivity that is lost in the direct link. We specify that we do not intend to resolve the internal structure of the ducting.

First, propagation is also defined by the refractivity gradient, not by the surface refractivity value.

Bending caused by reflection itself depends on the incident angle at ground level. Given a fixed geometric location (say R, local radius, at the surface), this angle depends on the impact parameter a=n*R. At a given impact parameter, a ray may propagate without reflection, or present reflection at several angles of incidence, depending on the **surface refractivity N** (not its gradient). The surface refractivity is resolved through direct paths as an accumulation of all the vertical gradients, except when a section is lost (ducting) and the chain of accumulation is incomplete. The reflection provides a different accumulation. We have added comments on this.

Moreover, being a simulation (or application of forward model, which does not model at all the propagation within a duct), the "ray" maybe cross the duct because the incident angle (the angle with the normal to the duct) is small enough and it is "refracted" into the duct. As a first thought, in this case one might expect bigger bending angles for the same impact parameter (or smaller impact parameters for the same bending). But, looking at the family of bending angle profiles in Figure 12, right panel, this is showing exactly the opposite. The second thought is one the incidence angle, which is decreasing with increasing N gradient (more bending towards the surface), thus the overall bending angle defined by equation 3 is becoming smaller and smaller because the second term in this equation is increasing. Is it correct? Could you please add a sentence on this?

We have added further explanations, in the main text and the figure. The curve bending/impact behavior described by the reviewer would be for direct propagation, for rays whose tangent point falls near the duct. The curve shown is at lower impact parameter, below the apparent horizon, and well below the elevation where the tangent point falls near the duct. See Fig 10.

- Page 12, row 13: if taking the fR from the interferogram shown for example in Figure 9, fD is always zero and thus fR = fI?

The fD is the Doppler of the direct signal, which is not zero. The interferogram is built to subtract the direct frequency fD to nearly zero. In the interferogram, the direct link appears as nearly zero by construction, as fR~=fD. This fD is obtained from the occultation data, following procedures which are a normal part of standard occultation processing (phase->Doppler->bending). fR is a smoothed version of fD, just to center fD in the time/frequency diagram.

If this is true, where is the fD taken from? From the measurements on the direct link (thus from the original carrier phase of the signal)?

Yes, the standard Doppler of the direct signal that would otherwise be evaluated, and which determines the impact/bending.

Taking simply the time derivative of the carrier phase? But it is well known that, in the lower troposphere and in particular in the sub- and tropical regions, atmospheric multipath often occurs. This means that the instantaneous (excess) Doppler components cannot be estimated by the carrier phase time derivative.

The reflected signal is visible before the direct signal reaches the low troposphere. Both may then be clear and distinct. As mentioned in the paper, in the lowest portion of the profile, the low atmosphere distorts both signals, and makes them non-separable. We do not assume that we can separate these, which is why there is a gap between the direct and reflected sections.

Wave optics techniques (FSI for example) should then be applied. Thus is the fD(t) for a particular time t when the fI(t) of the reflection is also present, computed with a wave optics approach? Could you please clarify this in the paper?

FSI, backpropagation, etc, are independent, and may if appropriately coded be used for this purpose. The presence of fR is a second signal that has followed an entirely different path (thus different impact/bending).As an example, should we mention here the work done by Gorbunov (to be published in the same special issue) https://www.atmos-meas-tech-discuss.net/amt-2017-189/amt-2017-189.pdf

- Page 16, row 20: Somehow doubting that the reflected beam will allow extracting information if the direct one is lost. This would be a "nice" case, but in reality the receiver will encounter low SNR in the lowest part of the atmosphere, is unable to track the direct beam, so why should the reflected one (which encountered even more of that lowest, variable part of the atmosphere) still be present?

Let us assume a descending occ. The reflected signal that we consider useful is observed **before** the direct is lost. The direct is clear because the TP is at higher altitude than the challenging low tropo, and the reflected is clear because it enters the low troposphere at a higher angle. Later both signals may be lost, probably at nearly the same time.

- General: it might be worthwhile to also think about corrections/removal that can be applied based on knowledge of the reflected beam. This could be very valuable for all the situations where the direct beam is already very noisy (and no reflected one is easily identifiable). Can you include a few words on this option?

Our approach was to have a supplementary set of measurements, at deeper impact parameters than the direct set of bending angles. The supplementary set should close an underdetermined set of direct measurements. Ducting is an example of obviously underdetermined set of direct bending angles.

- General: would be nice to include some general statement on what to expect from e.g. COSMIC-2 with its high gain occultation antennas.

We have added some final comments in the conclusion. It is of course always better. Not only for the antenna, but also if a wider bandwidth is recorded and made available (100 Hz sampling).

"Technical corrections"
Page 2, row 22: "either by polarization". This is not strictly true. The signal is right hand circularly (RHCP) polarized. The receiver antenna onboard the LEO must be sensitive to the RHCP polarization only, with a very low sensitivity to the cross-pol component (the left hand one). GPS receiver cannot separate the two components at all. The remaining energy in the LH component is something like a noise. In case of reflection with low incidence angles (_nadir reflections, typical for a GNSS-R receiver), the polarization of the reflected signal will be LH. Thus to make the receiver "sensible" to such signals, the antenna should be a LHCP one. Again the receiver is not separating anything.

We have restated to be more accurate. COSMIC, GRAS/METOP, or most existing RO receivers do not have the ability nor the intent to detect an LHCP component, even if it was there. The upcoming PAZ could. The sentence is intended to indicate that the reflection appears in the same polarization. It is also

intended to state that even if the receiver had the ability to record LHCP, (current space receivers don't) it would not be useful to separate both signals.

Page 2, row 26: what does it mean that the spectral distribution is compact? Maybe that the spectral components are within a small bandwidth? Use another adjective than compact...

It was "compact" as in "compact support", which here means indeed that it is nonzero only in a small bandwidth. We change for "narrow bandwidth".

Page 3, row 3: "signals" instead of "profiles"

Ok.

Page 3, row 4: "observations to be inverted to estimate the profile of..." instead of "measurements of the profile..."

Ok.

Page 4, row 2: it is certainly worth to cite ESA's plans on this, but maybe investigating is better than planning? And there is an Eart instead of Earth in previous line.

Ok.

Page 4, row 6: grazing "observations" instead of grazing "angles"

Ok.

Page 4, row 28: "by a PRN code" instead of "by the PRN".

Ok.

Page 4, row 33: what does it mean "orbital Doppler"? Is the Doppler experienced by the signal due to the relative radial movement between GNSS and LEO considering vacuum propagations? Please specify.

The Doppler that would be observed in a vacuum, yes. We rephrase this.

Page 5, row 1-2: what are you referring with "electromagnetic noise"?

It is perhaps better "measurement noise". The ensemble of all imperfections of the antennae, receiver, background noise, etc. In the process of encoding, transmission and decoding, we do not expect to retrieve exactly the original.

Page 5, row 3: again, is the Doppler subtraction referred to the "orbital Doppler" not better defined at Page 4, row 33? please clarify.

Yes. We fix that. There is a parenthesis explaining that by orbital Doppler we mean the straight-line propagation.

Page 5, row 4: what do you refer with "frequency broadening"? please clarify.

The emitted signal is a (modulated) very pure tone, with a very narrow frequency distribution. Interaction with the atmosphere does not just shift this frequency. The final tone is a sum of components shifted by different frequencies. The received signal is much wider. We clarify to explain that the spectrum of the received tone is broader than the narrow tone emitted.

Page 5, row 12: what about L2C tracking, which might be better than L1C/A?

Currently deployed receivers do not have that ability. Future receivers could benefit from its better correlation properties, and of dual-frequency tracking deep in the troposphere, including reflections. We do mention that thanks to L2C, dual frequency profiles could extend deeper, including reflections.

But the single item that would improve the ability to collect these data would be the ability to track or record both the direct and the reflected signal independently. They have an offset of some tens/few hundred Hz, and a delay that may add to several chip lengths. Current receivers track the reflected only because if falls within the bandwith, and only when the relative delay is less than one chip length (300m).

Page 5, row 20-21: this is the standard frequency beating phenomenon. Such details can be avoided.

Ok. We simplify the sentence, although following another comment that we received, we still mention that it is left in a small frequency band.

Page 6, row 22: you are naming the threshold you need to define simply as SVM (also later on in the paper) Could you please find another acronym/name to such threshold? SVM refers to "Support Vector Machine" which is a method, not a numerical value/threshold (see for example row 27 same page).

Ok. We have parsed the document to differentiate the algorithm, the output value, or the threshold applied to classify reflection presence/absence.

Page 7, row 22: reference to a supplementary movie. If you think it is ok, discard this comment. People will mostly download the paper and simply read it. Is it potentially worth to put the movie somewhere in an open repository (e.g., DropBox, Google cloud, ... ) and make reference to the repository? See also Page 8, row 29

The paper figures are already self-consistent, so the videos are not absolutely essential, although the videos are more complete and graphic. AMT allows supplementary material, available in the same location, which is probably better than Dropbox/etc. Besides AMT, this supplementary info (perhaps formatted differently) will also be deposited in IEEC's web page. However, AMT offers the guarantee of long-term storage. All in all, we prefer to keep it.

Page 8, row 10: I think that Ulaby could also be referenced well before. Probably in discussing coherency, or simply when you talking about electromagnetic reflection.

Ok. We have also added it to the introduction.

Page 8, row 28: Looking at Figure 7, I'd say that the fraction of reflections at lat > 45deg is always large, but between 45% and 65%. Unfortunately the figure uses grey levels, so it is not very clear. I suggest to use also different line styles, in order to better show the seasonal similitude (if I'm not wrong, in summer/winter there are always more/less reflections in both hemispheres). Maybe it is worth to point it in the paper also discussing figure 7.

Besides grey tones, it also uses points of several shapes. The same is also presented in the map in Fig 4 and videos. We expand on the explanation of the content in the caption and text.

Page 8, row 35: "opposite" instead of "contrary"?

Ok.

Page 11, row 33: Could you better define the 2D template used to cross-correlate the radio-holographic image?

We add an explanation in the text and in Figure 9. The reflection signatures are very similar, except a time reversal for rising/setting. It is a structure in the hologram at the expected frequency vs time to match a surface reflection. Can be obtained by average of several clear cases, but may also be done synthetically. The only purpose is to identify the interest time section within the occultation.

Page 12, row 5-9: Could you please specify as a further detail for a more generic audience why the reflected signal is always going from -25 Hz to 0 in an interferogram like the one shown in Figure 2? Or at least provide a reference for this (I guess that there is a theoretical explanation here : Beyerle, G., K. Hocke, J. Wickert, T. Schmidt, C. Marquardt, and C. Reigber, GPS radio occultations with CHAMP: A radio holographic analysis of GPS signal propagation in the troposphere and surface reflections, J. Geophys. Res., 107(D24), 4802, doi:10.1029/2001JD001402, 2002.)

We have elaborated more. Indeed, it may go beyond, and in Fig 2 there is an example of winding, but the receiver does not have wider bandwidth. And it is at negative frequencies, as the examples are setting occultations. We add also a reference there to Beyerle et al (2002), besides the reference that is already present elsewhere in the paper.

Page 12, row 19-21: this is unclear. What is this formal precision of the inversion? And where is the evidence that the "thickness" of the "line" is larger than the precision? Is this because no noise is visible? This is also repeated at page 14, row 10-11.

We deleted the mention to the thickness. We were underscoring that the wavy details of the curve are well above noise. We now just mention that the precision of the retrieved profile is very high, with details to be further explored.

Page 12, row 25-27: unclear. Are you inferring that atmospheric multipath is neglected (or rays are disentangled after some preprocessing)?

We just want to focus on reflections, as a different issue than multipath.
We assume that any standard procedure of GPSRO processing will handle multipath (of close paths) and diffraction, and can extract the standard direct profile. We thus focus on reflection as a different issue: signal received at the same time as the direct signal, but with an offset of several Hz.

Why are you talking about "diffraction" here? Diffraction from what? Earth's surface?

No. Propagation of a wave between 2 points is not a straight line, but a spindle of finite width. The Fresnel diameter for a 1.5 GHz signal propagating over 30000 km, is about a km wide. We are assuming that this is standard GPSRO processing. We thus add a reference (Kursinski et al. 2000).

Page 13, row 10: smaller typo

Ok.

Page 13; row 18: "by twice the incident angle" instead of "... grazing angle".

"Grazing" has been used in other works. We prefer to change to "elevation angle". Incident is with respect to the normal, and we mean its co-angle. The co-incidence angle would be correct, but it seems too odd.

Page 12, row 29-31: I guess that the wording "local spherically distributed" refracted field can be used instead of its explanation.

Ok.

Page 16, row 21: If you have a loss of track, you are loosing both the information carried by the direct and the reflected signal. Instead if I well understood, in the time series of observations, the reflected contributions to the bending angle characterizing the lower impact parameters come before the direct contribution. In this case it makes sense that in case the signal is lost and, if any, reflections are "tracked" they can be used to eventually fill in the gap. Not sure but if you have pure open loop data, you can always reconstruct both (if SNR allows it). Could you please better address this?

Ok. We reword it. We no longer mention here the loss of track, as this is not the main point, but rather adding information that is independent of the picture provided by direct data, which may be incomplete. To answer the reviewer, yes, the reflected data that fills a tracking void has been measured **before** the loss of track.

Page 16, row 24-27. Not clear. Please, rephrase it.

Ok.

Table 2, caption: "surfaces typologies". Again the reference to SVM value (see comment to Page 6, row 22)

Ok.

Figure 1, caption: GNSSR at higher elevation (strictly speaking in this case the elevation angle is smaller...)? What does it mean? I'd use nadir looking. Also, here you use GNSSR, while otherwise you use GNSS-R.

We try to reword. Higher than the 1 degree elevation in an occultation. We will use near-nadir. We also check the use of GNSS-R for consistency.

Figure 2, caption: "setting" instead of "descending".

Ok.

What does it mean "winding"?

It is related to aliasing, for a signal that is shifting in frequency, and rolls several times across the bandwidth (in the radiohologram it would look like turns in a coil). In the example shown, the signal is aliased only once. In other occultations, it crosses several times. Since the expression seems uncommon,

and it is minor, we modify the sentence use only the word "aliasing", and instead mention in the text that multiple aliasing may appear.

Figure 3: with the uncertainty of the tangent point being a few km, are land reflections over small islands truly land ones, or could also be sea ones? Maybe add a pre-caution.

No, coastal areas are difficult for precisely this reason. We mentioned this to some extent with polar land/ocean. We now add further caveats for coastal regions in general (Section on NWP value).

Figure 3, 4: it seems the Arctic is neither land nor sea. Are there no reflections observed there?

Most occultations present reflections, in fact. However the number of occultations per pixel is small to calculate the statistics. We now mention that in the caption.
Because the GPS satellites are not in a polar-like orbit, there are fewer events over extreme latitudes (see figure below: histogram of the latitude of the COSMIC RO events in a ~10 years time span in green, and those with SVM>0.5 in red—likely reflection)

[Figure]

At polar latitudes there are much less events, so when grouped in monthly steps and 2D cells (as in figures 3 and 4), the amount of event per cell and month is too low to be statistically significant in most of these latitudes.

Figure 4, caption. You are referring to COSMIC here. Not sure in the text this is made. And, for instance, it is not made in Fig 3, caption, but it is made in the caption of Figure 7. Again also here is made reference to the "supplementary movie.."

It was in page 5 L8. We parse the document to underscore that this was done with COSMIC profiles. In fact this was mentioned (now more explicitly) in the general description of reflections. The figure is otherwise self-contained. The supplementary movie shows the same seasonal info already shown in the figure, only more graphically.

Figure 8: at 50% ref, there seems to be a vertical line for all cwv, is this some threshold trigger?

We notice this (and other) vertical structures, but after checking we do not find a "trigger". 50% is a common value at midlatitudes, where the column of water vapor changes over the months from dry to wet, spreading the distribution vertically. We do not draw any conclusion, but do not find any issue either.

Figure 10: might be better in impact height (IP - RoC)

We considered that. However, it does not help visually very much: IP-RoC has an apparent elevation of about 2 km at low altitude, so the reflection also seems to appear "floating" in IP-RoC space.

Figure general: they need to be made more coherent, font sizes are sometimes very large, sometimes okay, sometimes small. Units missing, etc.

We parse them to improve where possible, and have specified units in the caption if needed. As for font size, many figures are intended as single-column, and appear too large only when plot as full page (in draft mode, presumably will become single column in final).

---

## Author Comment (AC2) · 14 Nov 2017

The paper presents an overview of the detection of reflected signals in radio occultation events and discusses the potential use of additional information that could be derived from such data. It is well readable, also covers some basics and appears complete, although the authors could sometimes be more concise. Some statements and claims are made that should be clarified or corrected, see my comments below.

To summarize, the paper is acceptable for publication after addressing the following issues.
- Page 2, line 20: "Also, at this very low elevation, and unlike at higher incidence angles, a reflection does not lead to a reversal of the polarization."
I think the description of reflection off an infinite surface by use of Maxwell's equations does not entail a dependence of the polarization from the incidence angle as claimed above. The authors need to either prove this claim for the geometry at hand or give some reference.

At normal incidence, reflection of a circularly polarized signal switches handedness (the EM field keeps rotating in the same orientation, but propagation reverts). Since GPS is right-hand circularly polarized, it will switch to left (again, at normal incidence). We add a reference to a section of Born & Wolf and Ulaby for reflection and polarization as a function of the incidence angle. We add later another (Zavorotny & Voronovich, 2000)

- Page 4, line 28: "GPS satellites carry atomic clocks, which produce pure sinusoidal tones. Before emission, these tones are modulated by characteristic bitstreams. ..." Unless I misunderstand things, the GPS carrier signal is primarily generated by some (highly stable) oscillators for the Lx frequencies, whose signal is modulated with information derived from the atomic clocks, plus ancillary data. There exist different technologies for atomic clocks (e.g. Cs or Rb based), whose details are of little interest as far as the emission, reception, and further processing of the carrier signal is concerned. Since this is also probably not relevant for this work, I recommend that the authors correct or remove this claim there and in other places of the manuscript.

Since the details are irrelevant, we simplify to "GPS satellites produce very pure sinusoidal tones. Before emission, these tones are modulated by characteristic bitstreams. ...". We still mention the bitstream, as the receiver cannot listen directly to the pure L-band tone. The received "tone" is built by demodulation.

- Page 6, line 27: "... radio occultation events with SVM value greater than 0.25 are identified as reflections. " The use of the term "SVM value" is probably some slang. A couple lines above the term "scalar" was used. Further down, "... SVM threshold." I would recommend to use a more coherent term for the output variable of the SVM.

Ok. We have parsed the manuscript to verify accuracy of the expressions. SVM (Support Vector Machine) is an algorithm. We have changed as appropriate to "value" when we refer to the output of this algorithm, and "threshold" for the value where we assume that a reflection has been observed.

- Page 8, line 35 to page 9, line 3: "These patterns, consistently associated with geographic and seasonal features, do not suggest a direct relationship with instrumental problems or performance. They may, at most, be linked to instrumental performance issues if these arise under certain geophysical conditions related to seasons or geography."
I am not sure what the authors wanted to tell here. Is there evidence for variations of instrumental performance, for fundamental instrumental limitations, or other? Please rephrase.

We have rephrased. We do not suspect that instruments vary in performance. Since there is a clear geographic and seasonal association (as opposed to random), of the presence/absence of a reflection signature in individual profiles, we state that it is unlikely that this presence/absence is due to some kind of unidentified instrumental issue. Therefore we attribute it to properties of the atmosphere and ocean.

- Page 9, line 7: "... The fields from the ECMWF ERA Interim analysis are used here to determine the weather state. The correlations presented in the following are all computed using monthly averages evaluated on cells of 10_x10_ over the oceans, with the reflection fraction compared against several environmental variables."
How useful is the use of monthly averages for determining the correlation of the fraction of reflections with several atmospheric variables, or quantities derived from the weather situation? For "slow" variables such as SST this is certainly fine. But given the large cross-correlation for some of the quantities given in table 3, one might be asking if using the actual situation instead of some average might lead to different results. Did
the authors investigate this?

Yes. With this volume of data, correlating the individual data (several million profiles) or clustered subsets (several thousand clusters) leads to very similar results. We comment now that.

- Page 9, line 32: "... must not be directly linked ..." Should "must not" rather read "cannot"?

Ok.

- Page 10, line 6: "As mentioned above, the significant sea wave height is nearly uncorrelated (r=0.04), although wind speed over sea has a moderate positive correlation (0.43). This was somewhat unexpected, as stronger winds correspond to rougher sea surfaces, which could seem to link to less chances of coherent reflections."
Not sure if it is relevant, but for measurements of wind speed over sea (e.g. scatterometry and altimetry), instead of SWH the correlation with smaller scale features of the sea surface such as capillary waves is exploited. These are important to the description of reflection and damping of (Ku/C band) signals at larger incidence angle.

Theoretical considerations (Page 9, L11) already indicated that at this incidence angle and in the L band, the sea surface is specular even for large waves (even more for capillary waves). That low correlation with SWH was not unexpected.
The interesting part with the wind is that it is a positive correlation (better specular reflection with wind), whereas standard scatterometry (closer to normal incidence!) is based on the opposite: a decrease in specular reflection and increase in diffuse scattering with wind. This must be a different physical link with wind (indeed, we later point to atmospheric mixing as the link).

- Page 11, line 15: "that the reflection flag, either a qualitative present/absent, or the quantitative SVM value, stems from the observation, and that the knowledge that an occultation is expected a priori to show lower OMB difference than an average occultation is already a supplemental NWP value to the standard profiles of direct non-reflected signals."
Did anybody already see positive impact from using that information? The paper by Healy says that this appears to be difficult in practice. So why is it "already a supplemental NWP value"?

We have rephrased this. No, it has not yet been tested in an NWP environment. The sentence is supposed to mean
  1) that the SVM analysis is providing information (a modulation of the apriori error estimation for non-reflected) that is generally useful in NWP assimilation of any kind of data.
  2) "already" because the paper presents a second reflection product: an extension of the bending angle profile, which is in addition to the SVM.

- Page 12, line 19: "The formal precision of the inversion ..." Can the authors please explain what they mean by "formal precision"?

We have rephrased. The inversion is a fit, and provides an error estimate of the fit ("formal precision"), which is a good measure of the precision of the result, although may not be a good estimate of the accuracy.

- Page 13, line 11: "This is a physically new phenomenon that must be included ..." I don't think that reflection is a new phenomenon, it is just that it needs to be taken into account. Recommendation: discard "physically new".

We rephrased. It is different from what is involved in GNSS occultations (i.e. only refraction).

- Page 13, line 18: "Due to reflection, the direction of the ray suddenly changes at the surface, ..."
"suddenly"? Use a less prosaic description of trivial reflection.

Rephrased for a simpler sentence.

- Page 14, line 12: "Interestingly, the slope of the reflected profile is very sharp, when compared with the direct profile. ..."

When looking at eq.(3) and taking the derivative w.r.t. a, one has two contributions of different origin. The second term on the r.h.s is purely geometric and thus more or less trivial, while the first one involves a derivative of the kernel K of eq.(4) that would be quite interesting to see a discussion about.

We further discussed about the general behavior of the reflected bending, and this strong derivative. This derivative wrt a is largely dominated by the second term (geometric), as the kernel does not have a peak (large derivative) for reflected rays.

- Page 14, line 17: "but the fact of having reached the surface, ..."
It is actually an assumption here that reflection is off the (Earth's) surface, and not some reflecting layer, isn't it?

The observed Doppler of the "reflected" signals indicates in general that this layer is not very far from the surface. But we have added the comment, notably in the conclusion, that there is some possibility that some elevated layer causes the reflection in some cases, and that a user should verify if the apparent reflection is compatible with the surface. We consider this a case that is best handled through background check.

- Page 14, line 24: "For direct, non-reflected propagation paths, this dependency of the properties of the bending with respect to the atmosphere, presents sharp peaks at the respective tangent altitudes."
The kernel K has an (integrable) singularity, not just a "sharp peak". The following discussion on that page and also fig.11 is based on the naive assumption of some "sharp peak" and needs to be corrected.

We have rephrased for clarity.
We do not "correct" anything, nor find any problem with this integrable singularity. We mention the "peak" because the kernel of a direct path heavily concentrates the weight near the tangent point height, and is the basis for the high vertical resolution of radio occultations. This is not the case in a reflected path, where the distribution of weight vs altitude is much more uniform.

- Page 14, line 32: "The reflected kernels always include the entire atmosphere, and are always sensitive to the low troposphere. Among the direct paths, only a few are sensitive to the low troposphere, and some may be missing. The reflected kernels have therefore the ability to fill any section where direct data are not sufficiently sensitive." Is this "filling of missing sections" just some hope, or has it actually been done or shown? If not, remove or weaken the claim in the last sentence. Given the discussion in that paragraph and the first one on page 15, I recommend that strongly.

We add the word "potential". The example in Fig 12 does show that the reflected profile provides sensitivity to a ducting layer, in a simple case where the direct profile cannot. We specify in the conclusion that future work will be dedicated to practical use in a NWP context.

- Page 15, section 4.3: "Value of reflected data under superrefraction"
For the sake of completeness of the paper - and also for understanding this section – it would be good if the authors briefly explained superrefraction and ducting.

We have parsed the paper, as they had been used inaccurately. We add definitions for both.

- Page 15, line 17: "The additional refractivity is shaped as an error function, leading to a vertical gradient that may be strong. ..."
This needs to be explained better. From fig.12 I would expect that the vertical gradient is strongest near 2 km height, so the relation to surface refractivity is only through the property of the employed model (assumptions). The conclusions described there are not clear to me. What am I missing?

We have expanded this section. We explicit that this is a numeric experiment that demonstrates sensitivity of the reflected profile that fills a sensitivity gap of the direct profile.
The problem with a gap is not only a missing portion of the atmosphere. Between the gap and the surface, the direct profile may provide further information of the refractivity **gradient**, but only weak constraints on the refractivity (because a portion of the gradient is missing).
A reflected profile provides a closure with a **different** integral of the refractivity profile. This closure is not extremely good at any particular altitude, but being complete, adds the required constraint on the **refractivity below the gap.**

- Page 16, line 19: "Although the reflected section of the profile contains less independent information than the direct section, it contains a few unique capabilities. It can resolve voids in the direct profile, for instance in the event of loss of track."
Can one really resolve voids in the event of loss of track? Has this claim been demonstrated somewhere?

We have rephrased here (and in the paragraphs above).
1) Reflected data are observed at the same time as the direct scans the mid troposphere. If track is lost in the low troposphere, reflected data is a supplement. In the list of occultations, there are many events where the profile does not completely reach the surface, but a reflection is seen.
2) We are not trying to provide a refractivity profile with voids resolved. We are trying to produce useful constraints to the refractivity profile that are vertically complete and mostly involve the low troposphere, which is the weaknesses of the direct profiles.

Spelling:
- Page 18, line 11: "Marcquardt" -> Marquardt

Done

Running a spell checker over the entire text might find a couple typos.

Ok.

---

## Referee Report (RR1)

The paper presents a nice overview of the detection of reflected signals in GPS radio occultation events and discusses the additional information that could be derived from such data, including potential benefits from using this information in NWP. It is well organized and readable, also covers some basics and appears to be complete. Further work is encouraged, showing the benefit in actual NWP applications.

All in all, the paper may be accepted for publication after addressing the following issues.

- Page 2, line 32: "However, at this very low elevation, a reflection does not lead to a reversal of the polarization, and the reflected signal is still right-handed. Although GPS receivers are designed to separate in most cases the different received components, either by polarization, through the Doppler shift, or by delay through PRN modulation, the specific case of the direct and reflected paths near the horizon appears as particularly challenging."

I think the authors want to express that, for given electromagnetic properties of the reflecting surface, amplitude and phase of the reflected signal may be calculated. In general, it is a superposition of (linear or) circular polarized waves. The authors are probably interested only in the case where the incident angle is larger than Brewster's angle (for water), although they mention e.g. CYGNSS and GEROS-ISS where this is not the case. Furthermore, a receiver needs some kind of "hardware assistance" (antennae) to be able to discriminate signal polarization.

Please rephrase to make the author's intentions better understandable, or shorten appropriately, as the polarization is not really important here.

- Section 2, page 5-6: L1 and L2 are explained, but not P(Y) or L2C.

- Page 8, line 5: "... as well some cases of non-reflective interaction with the surface, such as a mirage."

Is it really an interaction with the surface, rather than some layer above?

- Page 9, line 24-27: "These patterns, consistently associated with geographic and seasonal features, do not suggest a direct relationship with any instrumental problem or performance. They may, at most, be linked to instrumental performance issues if these arise under certain geophysical conditions related to seasons or geography."

Either there is a contradiction here, or I do not understand what the authors mean by "performance" and "issues". Also, is there some simulation of occultation events which helps to understand where there is the real problem? Does it have to be the "instrument", or another part of the chain linking the primary event to the data being seen by the user? Instead of speculation, why not suggesting appropriate studies?

Please reformulate.

- Page 9, line 31: ERA Interim is mentioned, but a reference is missing. Consult the ECMWF website for a proper citation.

Furthermore, it should be better explained also in the main text how the ERA data are used in the calculation of the correlations, not only in the caption of table 3. For variables that are single-level and that would be considered "slow", like sea-surface temperature, this may be unimportant, while it may be different for others which are spatially and temporally varying faster (like relative humidity). The reader might ask whether averaging over most of the troposphere significantly affects the

conclusions.

- Page 16, line 5 and page 36, fig. 11: misrepresentation of singularities

The kernel $K(r)$ in eq.(4) has an integrable singularity at nr=a, not a
"narrow peak", it is smooth for nr>a, and it is undefined for nr<a.  It is
a density that depends on the measure, and by a suitable change of variable
the singularity may be dealt with.

At the same time, fig.11, claiming to show K, shows several lines with some
inappropriate peak.  This is wrong.  Please check the manual of the
plotting tool used how to properly plot singular functions.

Both needs to be corrected.

- Page 29, caption of fig.4: it should probably read "SVM output value".
  Furthermore, "... events per pixel is small to perform statistics ..."
  should probably read "... too small to derive sensible statistics".

- Page 33, fig.8: What is the units of CWV?

Spelling etc.:

  Some cases which might have been found by a tool:

- Page 12, line 3: "contitute".

- Page 17, line 3: "direct direct"

- Page 18, line 17: "sensitivivity"

---

## Author Response (AR2)

**Answers to reviewer 2:**

We have adopted most of the comments of the reviewer. We specify in the following. Reviewer's text is shown in blue, with our comments in black. For your convenience, please see the document with differences highlighted.

The paper presents a nice overview of the detection of reflected signals in GPS radio occultation events and discusses the additional information that could be derived from such data, including potential benefits from using this information in NWP. It is well organized and readable, also covers some basics and appears to be complete. Further work is encouraged, showing the benefit in actual NWP applications.

All in all, the paper may be accepted for publication after addressing the following issues.
- Page 2, line 32: "However, at this very low elevation, a reflection does not lead to a reversal of the polarization, and the reflected signal is still right-handed. Although GPS receivers are designed to separate in most cases the different received components, either by polarization, through the Doppler shift, or by delay through PRN modulation, the specific case of the direct and reflected paths near the horizon appears as particularly challenging."
I think the authors want to express that, for given electromagnetic properties of the reflecting surface, amplitude and phase of the reflected signal may be calculated. In general, it is a superposition of (linear or) circular polarized waves. The authors are probably interested only in the case where the incident angle is larger than Brewster's angle (for water), although they mention e.g. CYGNSS and GEROS-ISS where this is not the case. Furthermore, a receiver needs some kind of "hardware assistance" (antennae) to be able to discriminate signal polarization. Please rephrase to make the author's intentions better understandable, or shorten appropriately, as the polarization is not really important here.

We have rephrased the paragraph to be more accurate. We are describing the different hardware discriminations that are been applied (antenna polarization, correlator, receiver bandwidth), and how at very low elevation, a reflection, which was not the intended target, passes all of them and is captured.
We now begin by mentioning that that the hardware performs a number of discriminations before recording any data.
Polarization is one of these hardware discriminating layers. Mention to polarization, CYGNSS and GEROS is done by contrast, as these two are an example where polarization would discriminate between direct signals and reflections, whereas in an occultation it is not the case. As the reviewer mentions, this different ability to discriminate is associated to the incidence angle being well above/below Brewster's angle.

- Section 2, page 5-6: L1 and L2 are explained, but not P(Y) or L2C.

There is now a short description of both as modulation bitstreams.

- Page 8, line 5: "... as well some cases of non-reflective interaction with the surface, such as a mirage."
Is it really an interaction with the surface, rather than some layer above?

Modified to "near the surface".

- Page 9, line 24-27: "These patterns, consistently associated with geographic and seasonal features, do not suggest a direct relationship with any instrumental problem or performance. They may, at most, be linked to instrumental performance issues if these arise under certain geophysical conditions related to seasons or geography."
Either there is a contradiction here, or I do not understand what the authors mean by "performance" and "issues". Also, is there some simulation of occultation events which helps to understand where there is the real problem? Does it have to be the "instrument", or another part of the chain linking the primary event to the data being seen by the user? Instead of speculation, why not suggesting appropriate studies?
Please reformulate.

We have rephrased. We considered valuable to describe the main trends identified, over land and ocean, including a general inhibition in the "tropics", but with further seasonal and geographical modulations, and to narrow it down to more specific situations (we positively rule out that modulation is due to tracking, and present correlations with some meteorological situations to provide further indications). We do not consider this modulation as closed, and now explicitly encourage further studies on its physical mechanism, both here and in the conclusion.

- Page 9, line 31: ERA Interim is mentioned, but a reference is missing.
Consult the ECMWF website for a proper citation.

We add a reference (Dee et al, 2011).

Furthermore, it should be better explained also in the main text how the ERA data are used in the calculation of the correlations, not only in the caption of table 3. For variables that are single-level and that would be considered "slow", like sea-surface temperature, this may be unimportant, while it may be different for others which are spatially and temporally varying faster (like relative humidity). The reader might ask whether averaging over most of the troposphere significantly affects the conclusions.

We add further details in the main text, and an explanation why averaging "fast" fields is here not critical (because the inhibition of reflections is a large effect, it can be associated with the core of the distribution of those fields, not only with the extremes).

- Page 16, line 5 and page 36, fig. 11: misrepresentation of singularities
The kernel K(r) in eq.(4) has an integrable singularity at nr=a, not a "narrow peak", it is smooth for nr>a, and it is undefined for nr<a. It is a density that depends on the measure, and by a suitable change of variable the singularity may be dealt with.
At the same time, fig.11, claiming to show K, shows several lines with some inappropriate peak. This is wrong. Please check the manual of the plotting tool used how to properly plot singular functions. Both needs to be corrected.

Modified Eq 4 to explicit that K(r) is zero for nr<a. This is also mentioned in the figure. The figure has been updated, and the asymptotes are shown as distinct dotted lines.

- Page 29, caption of fig.4: it should probably read "SVM output value".

Ok. We also now explicit mention "fraction of occultation events identified as showing a reflection", in both the land and ocean figures. The mentions to the supplement movies, both in the text and in the captions, indicate "SVM output value".

Furthermore, "... events per pixel is small to perform statistics ..."
should probably read "... too small to derive sensible statistics".

Ok.

- Page 33, fig.8: What is the units of CWV?

Added (it was kg/m^2).

Spelling etc.:
Some cases which might have been found by a tool:
- Page 12, line 3: "contitute".
- Page 17, line 3: "direct direct"
- Page 18, line 17: "sensitivivity"

All corrected (plus a couple others).

**Information content in reflected signals during GPS Radio Occultation observations**

Josep M. Aparicio[1,3], Estel Cardellach[2,3], and Hilda Rodríguez[2,3]

[1]Meteorological Research Division, Environement and Climate Change Canada (ECCC), 2121 Transcanada Hwy, Dorval, QC, Canada
[2]Institut de Ciències de l'Espai (ICE), Consejo Superior de Investigaciones Científicas (CSIC), Cerdanyola del Vallès, Spain
[3]Institut d'Estudis Espacials de Catalunya (IEEC), Cerdanyola del Vallès, Spain

*Correspondence to:* Josep M. Aparicio (Josep.Aparicio@canada.ca)

**Abstract.** The possibility of extracting useful information about the state of the lower troposphere from the surface reflections that are often detected during GPS radio occultations (GPSRO) is explored. The clarity of the reflection is quantified, and can be related to properties of the surface and the low troposphere. The reflected signal is often clear enough to show good phase coherence, and can be tracked and processed as an extension of direct non-reflected GPSRO atmospheric profiles. A profile of bending angle vs. impact parameter can be obtained for these reflected signals, characterized by impact parameters that are below the apparent horizon, and that is a continuation at low altitude of the standard non-reflected bending angle profile. If there were no reflection, these would correspond to tangent altitudes below the local surface, and in particular below the local mean sea level. A forward operator is presented, for the evaluation of the bending angle of reflected GPSRO signals, given atmospheric properties as described by a Numerical Weather Prediction system. The operator is an extension, at lower impact parameters, of standard bending angle operators, and reproduces both the direct and reflected sections of the measured profile. It can be applied to the assimilation of the reflected section of the profile as supplementary data to the direct section. Although the principle is applicable also over land, this paper is focused on ocean cases, where the topographic height of the reflecting surface, the sea level, is better known apriori.

*Copyright statement.* ©Her Majesty the Queen in Right of Canada, as represented by the Minister of Environment and Climate Change Canada, 2017.

[revised manuscript text omitted]